# Learning to Summarize by Learning to Quiz: Adversarial Agentic Collaboration for Long Document Summarization

**Weixuan Wang**[1][†]    **Minghao Wu**[2][†]    **Barry Haddow**[1,3]    **Alexandra Birch**[1,3]

[1]School of Informatics, University of Edinburgh
[2]Tongyi Lab, Alibaba Group
[3]Aveni
{weixuan.wang, bhaddow, a.birch}@ed.ac.uk
minghao.wu@alibaba-inc.com

## Abstract

Long document summarization remains a significant challenge for current large language models (LLMs), as existing approaches commonly struggle with information loss, factual inconsistencies, and coherence issues when processing excessively long documents. We propose SummQ, a novel adversarial multi-agent framework that addresses these limitations through collaborative intelligence between specialized agents operating in two complementary domains: *summarization* and *quizzing*. Our approach employs summary generators and reviewers that work collaboratively to create and evaluate comprehensive summaries, while quiz generators and reviewers create comprehension questions that serve as continuous quality checks for the summarization process. This adversarial dynamic, enhanced by an examinee agent that validates whether the generated summary contains the information needed to answer the quiz questions, enables iterative refinement through multifaceted feedback mechanisms. We evaluate SummQ on three widely used long document summarization benchmarks. Experimental results demonstrate that our framework significantly outperforms existing state-of-the-art methods across ROUGE and BERTScore metrics, as well as in LLM-as-a-Judge and human evaluations. Our comprehensive analyses reveal the effectiveness of the multi-agent collaboration dynamics, the influence of different agent configurations, and the impact of the quizzing mechanism. This work establishes a new approach for long document summarization that uses adversarial agentic collaboration to improve summarization quality. In addition, we release the code to foster research along this line.[1]

## 1 Introduction

Summarization has become increasingly critical in modern natural language processing, as organizations and individuals face an ever-growing volume of textual information that requires efficient processing and comprehension (Gambhir & Gupta, 2017; Zhao et al., 2020; Zhang et al., 2021). Prior works in summarization have primarily focused on short to medium-length documents, where models can effectively capture the essential content and generate coherent summaries (See et al., 2017; Fabbri et al., 2019). Recently, there has been an increasing interest in long document document summarization, driven by the need to process extensive texts such as research articles, legal documents, and books (Huang et al., 2021; Kryscinski et al., 2022; Saxena & Keller, 2024).

Recent large language models (LLMs) have shown promising results for summarization tasks (Pu et al., 2023; Laban et al., 2023; Keswani et al., 2024). However, existing methods struggle with long documents, often leading to significant information loss, factual inconsistencies, and difficulty

---

[†] Equal contribution.
[1]https://github.com/weixuan-wang123/SummQ

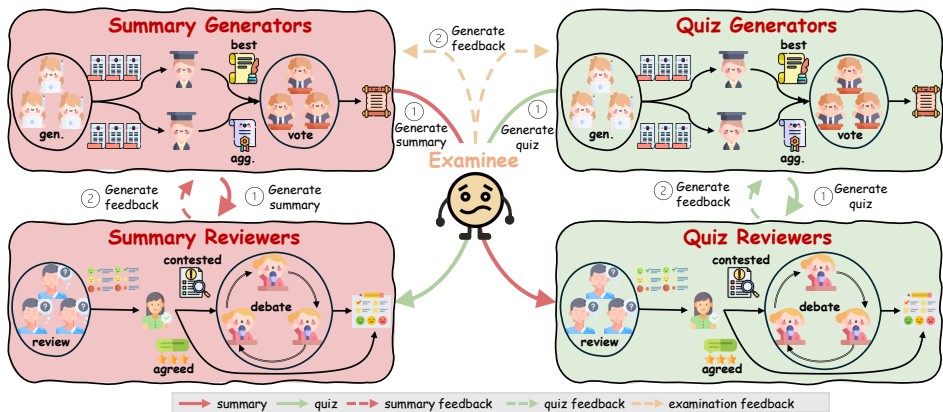

Figure 1: The overall framework of SUMMQ. It consists of two tasks, summarization and quizzing, and two types of agents: generator and reviewer, resulting in four groups of agents: *Summary Generators*, *Quiz Generators*, *Summary Reviewers*, and *Quiz Reviewers*. Additionally, we include an *Examinee* agent to check if quiz questions can be answered by the summary.

maintaining coherence across lengthy texts (Koh et al., 2023). Current approaches often fail to capture the nuanced relationships between distant parts of a document, resulting in summaries that may miss crucial information or introduce hallucinations (Chrysostomou et al., 2024; Tang et al., 2024a; Xia et al., 2024). Recently, multi-agent systems has demonstrated potential for improving complex reasoning tasks through collaborative interactions (Guo et al., 2024), yet their application to long document summarization remains underexplored (Fang et al., 2024; Kim & Kim, 2025).

To address these challenges, we propose SUMMQ, an adversarial multi-agent framework that leverages collaborative intelligence to generate high-quality summaries for long documents. Our approach divides specialized agents into two complementary task domains: *summarization* and *quizzing*. Within the summarization domain, we deploy summary generators that collaboratively create comprehensive summaries through independent drafting, aggregation, and collective voting, alongside summary reviewers that rigorously evaluate content quality through independent review and structured debate mechanisms. Simultaneously, the quizzing domain employs quiz generators to create comprehension quizzes that test the completeness and accuracy of generated summaries, while quiz reviewers ensure the quality, coverage, and appropriateness of these assessments. This dual-task framework creates a natural adversarial dynamic where the quiz generation process serves as a continuous quality check for summarization. The summary aims to provide comprehensive coverage of the document, enabling the quiz questions to be answered correctly, while the quiz challenges the information coverage, factuality, and coherence of the summary. Furthermore, an examinee agent is introduced to provide additional feedback, ensuring that the quiz questions can be accurately answered using only the generated summary. Through iterative refinement guided by multifaceted feedback, SUMMQ ensures that the final summaries are not only comprehensive and coherent but also factually accurate and verifiable.

To validate the effectiveness of SUMMQ, we conduct extensive experiments on three long document summarization tasks including MENSA (Saxena & Keller, 2024), BookSum (Kryscinski et al., 2022), and GovReport (Huang et al., 2021). Our results demonstrate that SUMMQ significantly outperforms existing state-of-the-art methods in terms of ROUGE (Lin, 2004) and BERTScore (Zhang et al., 2020), as well as LLM-as-a-Judge and human evaluations. Furthermore, more in-depth analyses highlight the effectiveness of our multi-agent framework in enhancing summary quality, the coverage of the generated quizzes, and the impact of various agent configurations.

Our contributions are summarized as follows:

- We introduce SUMMQ, a novel adversarial multi-agent framework that integrates summarization and quizzing tasks to enhance the long document summarization (see Section 3).
- We conduct comprehensive experiments on three long document summarization benchmarks, demonstrating that SUMMQ achieves state-of-the-art performance across multiple evaluation metrics and human assessments (see Section 4).

- We provide in-depth analyses of the multi-agent collaboration, the dynamics of the quizzing mechanism, and the impact of various agent configurations (see Section 5).

## 2 RELATED WORK

**Multi-Agent Systems** Recent advances in LLMs have enabled the development of multi-agent systems that harness the strengths of multiple agents to tackle complex tasks (Wang et al., 2024; Guo et al., 2024; Xi et al., 2025). These systems typically involve agent collaboration to boost performance, as seen in multi-agent debating (Du et al., 2024; Xiong et al., 2023; Chen et al., 2023a; Tang et al., 2024b) and discussion (Chen et al., 2024; Saha et al., 2024) for reasoning over short texts (Du et al., 2024; Tang et al., 2024b), paper review (Xu et al., 2023), dataset synthesis (Wang et al., 2025b), machine translation (Wu et al., 2024), and code generation (Huang et al., 2023; Wang et al., 2025a). Collaboration among agents introduces diverse perspectives and complementary skills, leading to higher-quality and more robust outputs.

**Long document Summarization** Long document summarization has seen various methods, including architectural optimizations (e.g., sparse attention (Liu et al., 2021; Ivgi et al., 2023; Bertsch et al., 2023), long-context finetuning (Beltagy et al., 2020; Guo et al., 2021), memory augmentation (Cui & Hu, 2021; Saxena et al., 2025), window extension (Press et al., 2021; Chen et al., 2023b; Su et al., 2024; Yen et al., 2024)) and chunking strategies (e.g., sliding window (Zaheer et al., 2020; Pang et al., 2023). LLMs with improved long-context abilities have shifted the field toward leveraging their strong language skills for summarization (Goyal et al., 2022; Ratner et al., 2023; Keswani et al., 2024), but still face challenges with context limits and maintaining coherence (Pu et al., 2023; Liu et al., 2023). Multi-agent systems have been explored to address these issues, enabling collaborative, more accurate summaries (Zhao et al., 2024; Fang et al., 2024; Jeong et al., 2025), though many still rely on self-verification, leading to biases and missed errors.

## 3 METHODOLOGY

In this section, we present SUMMQ for long document summarization, as illustrated in Figure 1. We first introduce the overall workflow in Section 3.1, and then describe the collaboration between the generator and reviewer agents in Section 3.2 and Section 3.3, respectively.

---

**Algorithm 1:** Overall SUMMQ Workflow

**Input:** Document $D$; Summary Generators $\mathcal{G}_s$; Quiz Generators $\mathcal{G}_q$; Summary Reviewers $\mathcal{R}_s$; Quiz Reviewers $\mathcal{R}_q$; Examinee $\mathcal{E}$; Number of iterations $T_{\text{iter}}$

**Output:** Accepted summary $S^*$, accepted quiz $Q^*$

1 $S^{(0)} \leftarrow \emptyset; Q^{(0)} \leftarrow \emptyset$ ;   // Initialize summary $S^{(0)}$ and quiz $Q^{(0)}$
2 **for** *iteration $t = 1$ **to** $T_{iter}$* **do**
3     $S^{(t)} \leftarrow \text{GENERATE}(\mathcal{G}_s, D, S^{(t-1)})$ ;   // Summary Generators produce candidate summaries
4     $Q^{(t)} \leftarrow \text{GENERATE}(\mathcal{G}_q, D, Q^{(t-1)})$ ;   // Quiz Generators produce candidate quizzes
5     $F_s^{(t)} \leftarrow \text{REVIEW}(\mathcal{R}_s, S^{(t)}, Q^{(t)}, D)$ ;   // Summary Reviewers produce feedback on summary
6     $F_q^{(t)} \leftarrow \text{REVIEW}(\mathcal{R}_q, Q^{(t)}, S^{(t)}, D)$ ;   // Quiz Reviewers produce feedback on quiz
7     $F_e^{(t)} \leftarrow \text{TAKEQUIZ}(\mathcal{E}, Q^{(t)}, S^{(t)})$ ;   // Examinee $\mathcal{E}$ takes the quiz based on the summary
8     $F_s^{(t)} \leftarrow F_s^{(t)} \cup F_e^{(t)}|_{\text{summary}}$ ;   // Merge feedback relevant to summary
9     $F_q^{(t)} \leftarrow F_q^{(t)} \cup F_e^{(t)}|_{\text{quiz}}$ ;   // Merge feedback relevant to quiz
10     **if** $F_s^{(t)} = \emptyset$ **and** $F_q^{(t)} = \emptyset$ **then**
11        **return** $(S^{(t)}, Q^{(t)})$ ;   // If no issues, accept and return the summary and quiz
12 **return** $(S^{(T)}, Q^{(T)})$

---

### 3.1 OVERALL WORKFLOW

The overall workflow of SUMMQ, depicted in Figure 1, involves two primary tasks: summarization and quiz generation, supported by two types of agents: generators $\mathcal{G}$ and reviewers $\mathcal{R}$. These tasks and agents combine to form four distinct components: *Summary Generators* $\mathcal{G}_s$, *Quiz Generators* $\mathcal{G}_q$, *Summary Reviewers* $\mathcal{R}_s$, and *Quiz Reviewers* $\mathcal{R}_q$. Additionally, an *Examinee* agent $\mathcal{E}$ is incorporated to validate that the quiz questions can be accurately answered using the summary.

The interaction between summarization and quiz generation creates a natural adversarial framework that continuously improves summarization quality. In this framework, the summary aims to provide comprehensive coverage that enables correct answers to quiz questions, while the quiz challenges the information coverage, factuality, and coherence of the summary. This dual-task approach ensures that both components evolve together, resulting in summaries that are not only informative but also verifiable through targeted questioning.

The iterative workflow of SUMMQ, as detailed in Algorithm 1, operates through a systematic process of generation, reviewing, and refinement. Beginning with an input document $D$, the system initializes empty summary and quiz states and enters an iterative loop for up to $T_{iter}$ iterations. In each iteration $t$, the process unfolds in four key stages. First, the Summary Generators $\mathcal{G}_s$ produce a candidate summary $S^{(t)}$ based on the document and any previous summary, while Quiz Generators $\mathcal{G}_q$ simultaneously generate a candidate quiz $Q^{(t)}$. Second, the reviewing phase begins with Summary Reviewers $\mathcal{R}_s$ evaluating the generated summary against both the quiz and original document to produce feedback $F_s^{(t)}$, and Quiz Reviewers $\mathcal{R}_q$ assessing the quiz quality to generate feedback $F_q^{(t)}$. Third, an Examinee agent $\mathcal{E}$ attempts to answer the quiz questions using only the generated summary, providing additional feedback $F_e^{(t)}$ that is then merged with the respective summary and quiz feedback streams. Finally, the system performs an acceptance check: if both feedback sets are empty, the current summary and quiz are accepted and returned. This iterative refinement continues until either high-quality outputs are achieved or the maximum iteration limit is reached. Note that GENERATE() and REVIEW() are detailed in Section 3.2 and Section 3.3, respectively.

This comprehensive workflow design ensures allows for continuous improvement based on concrete feedback, while the dual-task approach of simultaneous summary and quiz generation creates a natural consistency check that enhances the overall reliability and coherence of the final outputs.

---

**Algorithm 2:** GENERATE(): Generator Collaboration

**Input:** Document $D$; Previous summary/quiz $z'$; Generator agents $\mathcal{G} = \{g_i\}_{i=1}^{n}$; Aggregator agent $A_{\text{Agg}}$; Ranker agent $A_{\text{Ranker}}$
**Output:** Final summary/quiz $z^*$

1 **Phase 1: Independent Draft Generation**;
2 **for** *each generator agent $g_i \in \mathcal{G}$* **do**
3 $\quad \lfloor\ z_i \leftarrow \text{DRAFT}(g_i, D, z')$ ;   // Generate independent draft summary/quiz
4 $\mathcal{Z} = \{z_1, z_2, \ldots, z_n\}$ ;   // Set of all draft summaries/quizzes
5 **Phase 2: Draft Aggregation**;
6 $z_{\text{agg}} \leftarrow \text{AGGREGATE}(A_{\text{Agg}}, \mathcal{Z})$ ;   // Aggregate drafts into an unified summary/quiz
7 **Phase 3: Best Draft Selection**;
8 $z_{\text{best}} \leftarrow \text{BESTSELECT}(A_{\text{Ranker}}, \mathcal{Z})$ ;   // Select best individual draft
9 **Phase 4: Collective Voting**;
10 $\mathcal{C} \leftarrow \{z_{\text{agg}}, z_{\text{best}}\}$ ;   // Candidate summaries/quizzes for voting
11 **for** *each agent $g_j \in \mathcal{G}$* **do**
12 $\quad \lfloor\ \text{vote}_j \leftarrow \text{PREFER}(g_j, \mathcal{C}, D)$ ;   // Agent $g_j$ votes for preferred candidate
13 $z^* \leftarrow \arg\max_{z \in \mathcal{C}} |\{j : \text{vote}_j = z\}|$ ;   // Select the candidate with the most votes
14 **return** $z^*$

---

**Quiz Generation** The Quiz Generators $\mathcal{G}_q$ are responsible for producing a diverse range of question types, including multiple-choice, true-false, and short-answer questions. Through collaboration

among multiple Quiz Generators, the system generates 10 questions for each type as well as the corresponding answers, resulting in a total of 30 question-answer pairs per quiz.

## 3.2 GENERATOR COLLABORATION

The generator collaboration in SUMMQ is built around a multi-phase process that combines the strengths of multiple generator agents. As shown in Algorithm 2, this process unfolds in four phases:

**Phase 1: Independent Draft Generation**    The process begins with each generator agent $g_i \in \mathcal{G}$ working independently to create its own draft summary/quiz $z_i$ from the input document $D$ and any previous summary or quiz $z'$. This parallel approach naturally leads to diverse initial drafts, since different agents may emphasize various aspects of the document. We collect all these draft summaries or quizzes into a set $\mathcal{Z} = \{z_1, z_2, \ldots, z_n\}$, where $n$ denotes the total number of generator agents involved in the process.

**Phase 2: Draft Aggregation**    In the second phase, an aggregator agent $A_{\text{Agg}}$ combines the individual draft summaries or quizzes into a unified summary or quiz $z_{\text{agg}}$. This agent selectively combines the strengths of each draft and incorporates complementary information that individual agents may have overlooked. By drawing upon the collective knowledge across all drafts, $A_{\text{Agg}}$ creates a more comprehensive summary or quiz that harnesses the diverse perspectives and insights.

**Phase 3: Best Draft Selection**    Concurrently with the aggregation process, a ranker agent $A_{\text{Ranker}}$ evaluates each of the individual draft in $\mathcal{Z}$ to identify the highest-quality draft $z_{\text{best}}$. This parallel selection process serves as an important safeguard, ensuring that when a particular agent produces an exceptionally strong summary or quiz, it remains visible and is not overshadowed by $A_{\text{Agg}}$.

**Phase 4: Collective Voting**    In the final phase, we bring together the collective wisdom of all generator agents to make the ultimate decision between two candidates: the aggregated summary/quiz $z_{\text{agg}}$ and the best individual draft $z_{\text{best}}$. Each generator agent $g_j$ carefully evaluates both candidates from the set $\mathcal{C} = \{z_{\text{agg}}, z_{\text{best}}\}$ and casts their vote for the one they believe best captures the essence of the original document. The final summary/quiz is the candidate that receives the most votes.

---

**Algorithm 3:** REVIEW(): Reviewer Collaboration

**Input:** Document $D$; Summary/Quiz $z$; Reviewer agents $\mathcal{R} = \{r_i\}_{i=1}^{n}$; Number of debate rounds $T_{\text{debate}}$
**Output:** Decision $dec \in \{\text{ACCEPT}, \text{REJECT}\}$; Issue list $\mathcal{I}$

1 **Phase 1: Independent Reviewing** ;
2 **for** *each reviewer $r_i \in \mathcal{R}$* **do**
3     $\mathcal{A}_i \leftarrow$ ANNOTATE$(r_i, z, D)$ ;           // Review and annotate the summary/quiz
4 **Phase 2: Issue Categorization** ;
5 $\mathcal{M} \leftarrow \{a \mid a \in \bigcup_{i=1}^{n} \mathcal{A}_i \text{ and } |\{i : a \in \mathcal{A}_i\}| \geq 2\}$ ;       // Agreed issues
6 $\mathcal{C} \leftarrow \{a \mid a \in \bigcup_{i=1}^{n} \mathcal{A}_i \text{ and } |\{i : a \in \mathcal{A}_i\}| < 2\}$ ;       // Contested issues
7 **Phase 3: Contested Issue Validation via Debate** ;
8 $\mathcal{K} \leftarrow \emptyset$ ;               // Initialize valid issues
9 **for** *each contested issue $c \in \mathcal{C}$* **do**
10     **for** *debate round $t = 1$ **to** $T_{\text{debate}}$* **do**
11        **for** *each reviewer $r_i \in \mathcal{R}$* **do**
12           ARGUE$(r_i, c, D, z)$ ;   // Debate validity of issue $c$ with evidence from $D$ and $z$
13     $\text{vote}_c \leftarrow$ MAJORITYVOTE$(\mathcal{R}, c)$ ;       // Vote on whether issue $c$ is valid
14     **if** $\text{vote}_c = \text{VALID}$ **then**
15        $\mathcal{K} \leftarrow \mathcal{K} \cup \{c\}$ ;       // Keep valid contested issue
16 **Phase 4: Final Decision** ;
17 $\mathcal{I} \leftarrow \mathcal{M} \cup \mathcal{K}$ ;  // Combine major issues and kept contested issues into the final issue list
18 **return** $\mathcal{I}$ ;              // Return issue list

---

### 3.3 REVIEWER COLLABORATION

The reviewer collaboration in SUMMQ takes a systematic approach to quality assessment, where multiple reviewer agents work together to thoroughly evaluate generated summaries/quizzes and catch potential errors. As shown in Algorithm 3, this reviewing process also includes four phases:

**Phase 1: Independent Reviewing** The reviewing process begins with each reviewer agent $r_i$ conducting an independent and comprehensive review of the generated summary/quiz $z$ against the original document $D$. During this phase, each reviewer meticulously examines the summary/quiz to identify various types of quality issues, including factual errors, omissions of important information, redundant content, and other potential problems. The reviewers produce individual annotation sets $\mathcal{A}_i$ that capture their unique perspectives and assessment criteria.

**Phase 2: Issue Categorization** Once all reviewers have completed their independent reviews, the system categorizes the identified issues based on the level of agreement among reviewers. Issues that are flagged by at least two reviewers are classified as agreed issues $\mathcal{M}$, indicating a strong consensus that these problems genuinely exist. Conversely, issues identified by fewer than two reviewers are categorized as contested issues $\mathcal{C}$, suggesting disagreements that require further discussion.

**Phase 3: Contested Issue Validation via Debate** For contested issues in $\mathcal{C}$, where initial reviewer agreement is lacking, SUMMQ employs a structured debate mechanism to determine their validity. Each contested issue $c$ undergoes $T_{\texttt{debate}}$ rounds of debate, where all reviewer agents $r_i \in \mathcal{R}$ engage in evidence-based argumentation using the original document $D$ and summary or quiz $z$ as supporting materials. During each debate round, reviewers present their reasoning for or against the validity of issue $c$. After the debate rounds conclude, all reviewers participate in a majority vote to determine whether the contested issue should be considered valid.

**Phase 4: Final Decision** In the final phase, the reviewer collaboration reaches its final decision by consolidating all validated issues. The system combines the agreed issues $\mathcal{M}$ from Phase 2 with the valid contested issues $\mathcal{K}$ from Phase 3 to form the comprehensive final issue list $\mathcal{I} = \mathcal{M} \cup \mathcal{K}$. This issue list is returned to guide subsequent iterations of the generation process, ensuring that identified problems are systematically addressed in future revisions.

## 4 EXPERIMENTS

### 4.1 EXPERIMENTAL SETUP

**Evaluation** We evaluate SUMMQ on the long document summarization task using the MENSA (Saxena & Keller, 2024), BookSum (Kryscinski et al., 2022), and GovReport (Huang et al., 2021) benchmarks. Following standard protocols, we assess the generated summaries using ROUGE-1 (R-1), ROUGE-2 (R-2), ROUGE-L (R-L) (Lin, 2004), and BERTScore-$F_1$ (BS$_{F_1}$) (Zhang et al., 2020).[2] We also employ LLM-as-a-Judge evaluations with GPT-4.1 and GPT-5 (detailed in Appendix A), and conduct human evaluations. A case study is presented in Appendix F.

**Baselines** We compare SUMMQ against strong baselines from three categories: (1) **Supervised Fine-Tuning**: TEXTRANK (Mihalcea & Tarau, 2004; Jeong et al., 2025), LONGT5 (Guo et al., 2021), U.FORMER (Bertsch et al., 2023), SLED (Ivgi et al., 2023), and CACHED (Saxena et al., 2025). (2) **Prompting**: Proprietary LLMs including GPT-4O, GPT-4.1, GPT-5, O3, and open-source models such as DEEPSEEK-R1 and QWEN3-32B. (3) **Multi-Agent Systems**: HM-SR (Jeong et al., 2025) and C.MULTILLM (Fang et al., 2024).

**Ours** In our experiments, we consider two configurations of our approach: **SUMMQ$_{\text{SOLO}}$**, where each component employs a single agent for efficient and straightforward deployment, and **SUMMQ$_{\text{COMBO}}$**, where each component leverages multiple agents in an ensemble manner to facilitate collaborative generation and review. By default, we use GPT-4O as the agent backbone for both

---

[2]BERTScore model: `bert-base-uncased`.

Table 1: Overall results given by different methods on `MENSA`, `BookSum`, and `GovReport`. The best results are highlighted in **bold**.

| | MENSA | | | | BookSum | | | | GovReport | | | |
|---|---|---|---|---|---|---|---|---|---|---|---|---|
| | R-1 | R-2 | R-L | $BS_{F_1}$ | R-1 | R-2 | R-L | $BS_{F_1}$ | R-1 | R-2 | R-L | $BS_{F_1}$ |
| **Supervised Fine-Tuning** | | | | | | | | | | | | |
| TEXTRANK | 34.37 | 4.60 | 12.84 | 48.10 | - | - | - | - | - | - | - | - |
| LONGT5 | 20.77 | 2.26 | 10.03 | 45.01 | - | - | - | - | - | - | - | - |
| U.FORMER | - | - | - | - | 36.70 | 7.30 | 15.50 | 51.50 | 56.60 | **26.30** | 27.60 | **68.20** |
| SLED | - | - | - | - | 38.90 | 7.50 | 15.80 | 52.40 | **57.50** | **26.30** | 27.40 | 66.90 |
| CACHED | - | - | - | - | 42.80 | 10.50 | 18.80 | 54.40 | 57.00 | **26.30** | **28.19** | 67.30 |
| **Prompting** | | | | | | | | | | | | |
| GPT-4O | 25.78 | 7.24 | 13.59 | 59.67 | 23.02 | 1.81 | 12.23 | 58.52 | 31.42 | 11.87 | 17.61 | 63.43 |
| GPT-4.1 | 30.31 | 8.36 | 15.39 | 55.01 | 23.05 | 5.54 | 11.47 | 58.12 | 40.84 | 12.96 | 19.13 | 62.95 |
| GPT-5 | 37.38 | 9.14 | 17.11 | 60.44 | 23.98 | 5.69 | 12.38 | 58.55 | 41.52 | 12.52 | 19.23 | 62.55 |
| O3 | 32.84 | 8.54 | 17.09 | 59.27 | 22.00 | 5.24 | 11.51 | 58.44 | 38.28 | 9.93 | 17.47 | 61.19 |
| DEEPSEEK-R1 | 27.63 | 7.66 | 14.82 | 56.82 | 18.86 | 4.69 | 9.71 | 55.85 | 35.42 | 9.58 | 16.66 | 61.07 |
| QWEN3-32B | 23.49 | 5.58 | 12.77 | 55.76 | 20.19 | 4.68 | 10.68 | 56.51 | 35.52 | 10.80 | 17.07 | 61.08 |
| **Multi-Agent Systems** | | | | | | | | | | | | |
| HM-SR | 34.26 | 9.74 | 13.46 | 60.22 | - | - | - | - | - | - | - | - |
| C.MULTILLM | - | - | - | - | - | - | - | - | 47.90 | - | 19.70 | - |
| SUMMQ$_{SOLO}$ | 39.30 | 9.70 | 17.12 | 61.84 | 33.33 | 8.35 | 15.47 | 60.41 | 48.71 | 17.26 | 21.21 | 65.21 |
| SUMMQ$_{COMBO}$ | **41.58** | **10.96** | **18.24** | **62.76** | **44.62** | **11.14** | **20.38** | **61.49** | 52.79 | 18.47 | 21.76 | 65.46 |

(a) SUMMQ$_{COMBO}$ vs. GPT-4O

(b) SUMMQ$_{COMBO}$ vs. O3

Figure 2: The comparison between SUMMQ$_{COMBO}$ and baselines judged by GPT-5 on `MENSA` during iteration, where there are three GPT-4O agents in each component of SUMMQ$_{COMBO}$.

SUMMQ$_{SOLO}$ and SUMMQ$_{COMBO}$, deploying three agents in each component for SUMMQ$_{COMBO}$ unless otherwise specified. Moreover, the number of iterations $T_{iter}$ is set to three for all configurations and the number of debate rounds $T_{debate}$ is set to one. Prompts used in SUMMQ are in Appendix G.

## 4.2 AUTOMATIC EVALUATION RESULTS

**SUMMQ$_{COMBO}$ achieves strong performance with notable improvements on challenging datasets.** Table 1 reports the automatic evaluation results of all methods on the `MENSA`, `BookSum`, and `GovReport` benchmarks. Our SUMMQ$_{COMBO}$ configuration demonstrates strong performance across all datasets, achieving the best results on `MENSA` and `BookSum` across all metrics. On `GovReport`, while some supervised fine-tuning baselines (U.FORMER, SLED, CACHED) achieve competitive or superior performance on specific metrics due to their large-scale task-specific training, SUMMQ$_{COMBO}$ still outperforms all prompting baselines and shows substantial improvements over the SUMMQ$_{SOLO}$ variant. Notably, SUMMQ$_{COMBO}$ yields the most significant improvements on the challenging `BookSum` dataset, where it substantially outperforms all baselines across all metrics. Furthermore, SUMMQ$_{SOLO}$ configuration also performs strongly, consistently surpassing prompting baselines. These results confirm the advantages of our proposed multi-agent summarization framework, particularly the SUMMQ$_{COMBO}$ configuration, in handling diverse long documents.

**LLM-as-a-Judge evaluation highlights the effectiveness of SUMMQ$_{COMBO}$.** Figure 2 presents LLM-as-a-Judge results, comparing SUMMQ$_{COMBO}$ to strong baselines on the `MENSA` benchmark over multiple iterations. LLM judges (GPT-5) compare summary pairs and select the superior one, with each subfigure showing the winning rate of SUMMQ$_{COMBO}$ against different LLM agents

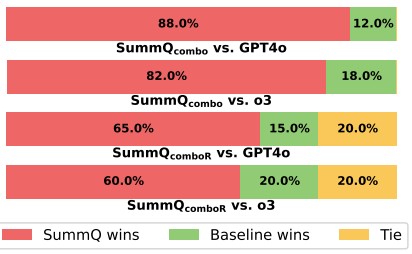

Figure 3: Human evaluation results comparing GPT-4O, O3, SUMMQ$_{\text{COMBO}}$, and SUMMQ$_{\text{COMBOR}}$.

Table 2: Results on MENSA obtained by SUMMQ, where one component contains multiple agents while other components contain only a single agent.

|  | R-1 | R-2 | R-L | BS$_P$ | BS$_R$ | BS$_{F_1}$ |
|---|---|---|---|---|---|---|
| SUMMQ$_{\text{SOLO}}$ | 39.30 | 9.70 | 17.12 | 62.19 | 61.55 | 61.84 |
| SUMMQ$_{\text{COMBO}}$ | **41.58** | **11.08** | **18.24** | **63.28** | 62.28 | **62.76** |
| **Only one component with 3 agents** | | | | | | |
| Quiz Rev. | 40.81 | 10.63 | 17.78 | 62.17 | 61.61 | 61.87 |
| Summary Rev. | 41.20 | 10.80 | 17.93 | 62.29 | 61.72 | 61.99 |
| Quiz Gen. | 40.40 | 10.80 | 17.95 | 62.54 | 61.72 | 62.11 |
| Summary Gen. | 40.72 | 10.96 | 18.07 | 62.70 | **62.39** | 62.53 |

Table 3: Results with different number of iterations $T_{\text{iter}}$ on MENSA with the SUMMQ$_{\text{COMBO}}$. All agents are GPT-4O.

| #iter. | R-1 | R-2 | R-L | BS$_P$ | BS$_R$ | BS$_{F_1}$ |
|---|---|---|---|---|---|---|
| 1 | 38.14 | 10.44 | 17.85 | 62.77 | 61.50 | 62.60 |
| 2 | 40.55 | 10.74 | 17.80 | 63.06 | 61.85 | 62.43 |
| 3 | 41.58 | 11.08 | **18.24** | **63.28** | **62.28** | **62.76** |
| 4 | **41.62** | **11.19** | 18.11 | 63.26 | 62.25 | 62.73 |
| 5 | 41.53 | 11.01 | 18.21 | 62.90 | 62.23 | 62.55 |

Table 4: Results with different number of agents in each component on MENSA with the SUMMQ$_{\text{COMBO}}$. All agents are GPT-4O.

| #agents | R-1 | R-2 | R-L | BS$_P$ | BS$_R$ | BS$_{F_1}$ |
|---|---|---|---|---|---|---|
| 1 | 39.30 | 9.70 | 17.12 | 62.19 | 61.55 | 61.84 |
| 2 | 40.49 | 10.20 | 18.02 | 62.23 | 62.11 | 62.16 |
| 3 | 41.58 | 11.08 | 18.24 | 63.28 | 62.28 | 62.76 |
| 4 | 41.81 | 10.94 | 18.30 | 62.82 | 62.46 | 62.34 |
| 5 | **42.52** | **11.49** | **18.56** | **63.53** | **62.99** | **62.96** |

(GPT-4O or O3), judged by GPT-5. Across all settings, SUMMQ$_{\text{COMBO}}$ consistently outperforms baselines, achieving higher winning rates and demonstrating the effectiveness and generalizability of SUMMQ$_{\text{COMBO}}$. The specific prompts and additional results judged by GPT-4.1 are in Appendix A.

### 4.3 HUMAN EVALUATION RESULTS

**Setup** We conduct a human evaluation using 20 randomly selected NLP papers published after June 2024, with five Ph.D. students as judges. Each judge compares two summaries considering *Informativeness*, *Coherence*, and *Factuality*. The performance is measured by the winning rate. To address length bias, we include both SUMMQ$_{\text{COMBO}}$ and a rephrased version SUMMQ$_{\text{COMBOR}}$ with shortened summaries. Details on the evaluation protocol and selected papers are in Appendix E.

**Results** Figure 3 presents the results of our human evaluation, comparing SUMMQ$_{\text{COMBO}}$ and SUMMQ$_{\text{COMBOR}}$ against strong baselines, including GPT-4O and O3. The results indicate that SUMMQ$_{\text{COMBO}}$ is preferred over GPT-4O and O3 with winning rates of 88% and 82%, respectively. Even after mitigating the potential length bias through rephrasing, SUMMQ$_{\text{COMBOR}}$ still outperforms GPT-4O and O3 with winning rates of 65% and 60%, respectively. These findings underscore the effectiveness of our collaborative multi-agent framework in generating high-quality summaries that are favored by human judges, even when accounting for differences in summary length.

## 5 ANALYSIS

**Multi-agent collaboration consistently excels for each component in SUMMQ.** We analyze each component of SUMMQ by replacing the single-agent setup with a multi-agent ensemble while keeping the other components as single-agent. As shown in Table 2, all components benefit from multi-agent collaboration, especially the *Summary Generators* and *Summary Reviewers*. These results highlight that collaboration and diverse perspectives significantly improve summary quality.

**More iterations of refinement does not always lead to better summaries.** We analyze the effect of varying the number of iterations $T_{\text{iter}}$ in SUMMQ (Algorithm 1) on summary quality for MENSA (Table 3). Performance generally improves from 1 to 3 iterations, with BERTScore-$F_1$ peaking at 62.76, but further iterations yield diminishing or negative returns. This suggests that too few

Table 5: Results given by SUMMQ$_{\text{COMBO}}$ with different LLMs as agent backbone on `MENSA`.

| | Proprietary Models | | | | | | | Open-source Models | | | | | |
| --- | --- | --- | --- | --- | --- | --- | --- | --- | --- | --- | --- | --- | --- |
| | R-1 | R-2 | R-L | BS$_P$ | BS$_R$ | BS$_{F_1}$ | | R-1 | R-2 | R-L | BS$_P$ | BS$_R$ | BS$_{F_1}$ |
| GPT-4O-MINI | | | | | | | DEEPSEEK-R1 | | | | | | |
| baseline | 26.61 | 6.56 | 13.77 | 57.66 | 55.87 | 58.15 | baseline | 27.63 | 7.66 | 14.82 | 54.98 | 58.86 | 56.82 |
| SUMMQ | 35.18 | 7.97 | 15.67 | 58.02 | 58.64 | 58.31 | SUMMQ | 30.71 | 7.77 | 15.04 | 61.99 | 58.77 | 60.30 |
| GPT-4.1 | | | | | | | QWEN3-32B | | | | | | |
| baseline | 30.31 | 8.36 | 15.39 | 56.00 | 54.12 | 55.01 | baseline | 23.49 | 5.58 | 12.77 | 55.86 | 55.99 | 55.76 |
| SUMMQ | 49.17 | 12.25 | 18.98 | 59.58 | 62.42 | 60.95 | SUMMQ | 26.50 | 5.80 | 13.14 | 57.13 | 59.09 | 58.02 |
| O3 | | | | | | | DEEPSEEK-R1-DISTILL-QWEN-32B | | | | | | |
| baseline | 32.84 | 8.54 | 17.09 | 57.38 | 59.81 | 59.27 | baseline | 26.66 | 6.35 | 13.77 | 58.83 | 55.57 | 57.07 |
| SUMMQ | 46.69 | 10.37 | 19.09 | 61.90 | 61.30 | 60.82 | SUMMQ | 31.07 | 6.99 | 14.70 | 58.80 | 57.15 | 57.83 |

iterations fail to fully leverage collaborative reasoning, while too many can introduce noise or over-refinement, indicating an optimal balance is needed.

**More agents lead to better performance, but with diminishing returns and increased cost.** As shown in Table 4, increasing the number of agents in each component of SUMMQ$_{\text{COMBO}}$ generally improves performance, but gains become smaller as more agents are added. For example, ROUGE-L rises from 17.12 (one agent) to 18.02 (two agents), but further increases yield only minor improvements. This highlights a trade-off between summarization quality and managing computational cost, as adding agents increases costs without proportional benefits.

**SUMMQ consistently achieves superior performance with diverse LLM agent backbones.** Table 5 shows that SUMMQ$_{\text{COMBO}}$ outperforms all baselines across various proprietary and open-source LLM backbones. Summarization quality strongly depends on the agent backbone: advanced models like GPT-4.1 and O3 perform better than their weaker counterparts. This highlights the importance of choosing robust LLMs to maximize multi-agent summarization performance. We also explore the impact of combining different LLMs within SUMMQ in Appendix B.

**Quizzing mechanism effectively improves the summarization quality.** To evaluate the contribution of the quizzing mechanism in SUMMQ, we ablate the quizzing mechanism in SUMMQ by removing generators and reviewers for quizzing and the examinee. As shown in Table 6, this leads to consistent performance drops across all metrics on `MENSA`. For SUMMQ$_{\text{SOLO}}$, R-1 drops by 5.00 and BS$_{F_1}$ by 6.40; for SUMMQ$_{\text{COMBO}}$, R-1 and BS$_{F_1}$ decrease by 2.09 and 2.17, respectively. These results show that the quizzing mechanism is crucial for enhancing summarization quality by comprehensively challenging the generated summaries.

Table 6: Results with and without quiz on `MENSA`.

| | R-1 | R-2 | R-L | BS$_P$ | BS$_R$ | BS$_{F_1}$ |
| --- | --- | --- | --- | --- | --- | --- |
| SUMMQ$_{\text{SOLO}}$ | 39.30 | 9.70 | 17.12 | 62.19 | 61.55 | 61.84 |
| ├ w/o quiz | 34.30 | 8.99 | 15.85 | 59.71 | 51.80 | 55.44 |
| SUMMQ$_{\text{COMBO}}$ | 41.58 | 11.08 | 18.24 | 63.28 | 62.28 | 62.76 |
| ├ w/o quiz | 39.49 | 9.33 | 17.13 | 61.10 | 60.39 | 60.59 |

**Quiz coverage gets more balanced across document segments as the iteration proceeds.** We divide each document into five equal segments and use GPT-4.1 to map quiz questions to segments. As shown in Figure 4, quiz questions initially focus on the beginning and end, which aligns well with the findings of Liu et al. (2024), but coverage becomes more balanced by iteration 3, with increased attention to middle segments. This shift indicates a more holistic document understanding, which is crucial for generating summaries that accurately reflect the entire content.

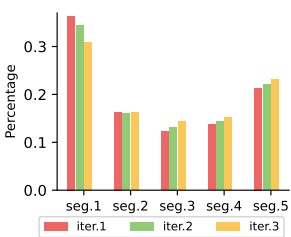

Figure 4: Quiz question distribution on `MENSA` with the SUMMQ$_{\text{COMBO}}$.

## 6 CONCLUSION

In this work, we introduce SUMMQ, a novel adversarial multi-agent framework that addresses critical challenges in long document summarization through collaborative intelligence between specialized summarization and quizzing agents. Our approach creates a natural adversarial dynamic where quiz generation serves as a continuous quality check, ensuring comprehensive coverage, factual accuracy, and verifiability of summaries through iterative refinement. Extensive experiments on three benchmarks demonstrate that SUMMQ achieves superior performance across multiple evaluation metrics including ROUGE, BERTScore, LLM-as-a-judge, and human assessments. Our comprehensive analyses reveal the effectiveness of multi-agent collaboration, the impact of the quizzing mechanism on summary quality, and the influence of various agent configurations.

## ETHICS STATEMENT

This work introduces SUMMQ, an adversarial multi-agent framework for long document summarization using large language models (LLMs). All experiments were conducted using publicly available datasets and LLMs, strictly adhering to their respective licenses and usage policies. No human subjects were involved in the development or evaluation of the system, except for the human evaluation, which was performed by consenting Ph.D. students with relevant expertise. We acknowledge that LLMs and datasets may contain inherent biases, which could affect the generated summaries and quiz questions. We encourage responsible use of our framework, with attention to fairness, transparency, and accountability in downstream applications.

## REPRODUCIBILITY STATEMENT

We are committed to reproducibility in this work. Detailed descriptions of the SUMMQ framework, including algorithms, agent configurations, and collaboration mechanisms, are provided in Section 3. Experimental setups, including model backbones, datasets, evaluation metrics, and baseline comparisons, are thoroughly described in Section 4. All datasets used are standard benchmarks, and references are included for accessibility. Prompts and implementation details are provided in Appendix. To further support reproducibility, we will release our code and experiment scripts upon publication, enabling other researchers to replicate and extend our results.

## THE USE OF LARGE LANGUAGE MODELS (LLMS)

In preparing this work, we utilize large language models (LLMs) as general-purpose tools to assist with writing polish and grammar correction. The LLMs are not involved in research ideation, experimental design, or substantive content generation. Their role is limited to improving the clarity and readability of the text, ensuring grammatical accuracy, and refining the presentation of our findings. All scientific contributions, analyses, and conclusions are solely the work of the authors.

## ACKNOWLEDGEMENT

This project has received funding from UK Research and Innovation (UKRI) under the UK government's Horizon Europe funding guarantee [grant numbers 10039436].

The computations described in this research were performed using the Baskerville Tier 2 HPC service (https://www.baskerville.ac.uk/). Baskerville was funded by the EPSRC and UKRI through the World Class Labs scheme (EP/T022221/1) and the Digital Research Infrastructure programme (EP/W032244/1) and is operated by Advanced Research Computing at the University of Birmingham.

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

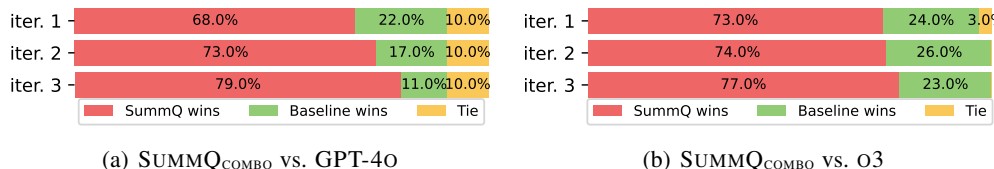

(a) SUMMQ_COMBO vs. GPT-4O               (b) SUMMQ_COMBO vs. O3

Figure 5: The comparison between SUMMQ_COMBO and baselines on `MENSA` judged by GPT-4.1 during iteration, where there are three GPT-4O agents in each component of SUMMQ_COMBO.

Table 7: Results with different combinations of LLMs in each component on `MENSA` with the SUMMQ_COMBO, ✓ indicates the LLM is used.

| GPT-4O | GPT-4.1 | O3 | GPT-5 | R-1 | R-2 | R-L | $BS_P$ | $BS_R$ | $BS_{F_1}$ |
|---|---|---|---|---|---|---|---|---|---|
| ✓✓✓ | | | | 41.58 | 11.08 | 18.24 | 63.28 | 62.28 | 62.76 |
| ✓ | ✓ | | ✓ | 48.82 | **13.79** | 21.46 | 63.32 | 64.66 | 63.97 |
| ✓ | ✓ | ✓ | | 48.06 | 13.17 | 21.44 | 62.63 | 64.25 | 63.42 |
| ✓ | | ✓ | ✓ | **49.42** | 13.26 | **22.90** | **63.71** | **65.17** | **64.42** |

# A LLM-AS-A-JUDGE EVALUATION

## A.1 LLM-AS-A-JUDGE SETUP

We employ an LLM-as-a-Judge evaluation framework to assess the relative quality of generated summaries. For each document, we compare pairs of summaries using the evaluation prompt shown below. To mitigate positional bias, we reverse the order of summaries for each pair, ensuring that each comparison is evaluated twice with alternating positions. The judge is instructed to determine which summary better meets the evaluation criteria or whether they are of equal quality.

---

**LLM-as-a-Judge 1: Evaluation Prompt**

*SYS PROMPT*:

You are an expert evaluator tasked with objectively assessing the quality of text summarizations.
Your response must strictly follow this format:
Reasoning: [Brief, precise explanation based on the criteria above.]
Better one or Equal: [Summary 1 or Summary 2 or Equal]

---

*USER PROMPT*:

Evaluate the following document and two summaries provided below. Determine which summary better meets the evaluation criteria provided, or whether they are equal.
Document: "{text}"
Summary 1: "{summary 1}"
Summary 2: "{summary 2}"

---

## A.2 ADDITIONAL RESULTS USING GPT-4.1 AS THE JUDGE

We provide additional LLM-as-a-judge results judged by GPT-4.1 in Figure 5, complementing the GPT-5 results in Figure 2. These results show that SUMMQ_COMBO consistently outperforms all baselines across iterations, further validating the effectiveness of our method.

## B    COMBINATION OF DIFFERENT LLMS

**Diverse agent combinations leverage complementary strengths.**    We have demonstrated the effectiveness of SUMMQ_COMBO with multiple GPT-4O agents, but we also explore the impact of combining different LLMs within our framework. Table 7 presents results for various strategies that mix agent types, including GPT-4O, GPT-4.1, O3, and GPT-5. The results show that ensembles composed of diverse LLMs generally outperform those using a single model type. These combinations leverage complementary strengths, such as distinct reasoning capabilities, knowledge domains, or summarization styles. For instance, the pairing of O3 and GPT-5 achieves the highest scores, likely due to O3's advanced reasoning and GPT-5's robust routing and selection abilities. This diversity enables the system to generate more comprehensive and higher-quality summaries, as agents can mitigate each other's weaknesses and collectively address a broader range of content and quality challenges.

## C    ACCURACY OF QUESTIONS EVOLVE

**Quiz answer accuracy improves throughout iteration.** As the iteration proceeds, the accuracy of answering quiz questions based on the generated summaries improves steadily. Figure 6 illustrates that all the question types exhibit this upward trend: multiple-choice (MC) and true-false (TF) questions achieve higher accuracy more rapidly, while short-answer (SA) questions, which demand deeper comprehension, show a more gradual improvement. This pattern underscores that agentic collaboration and iterative refinement in SUMMQ enhance summary quality, as reflected in the improved performance on increasingly complex quiz questions.

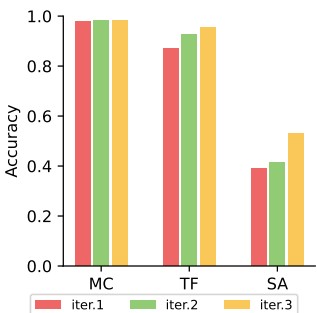

## D    COST ANALYSIS

Figure 6: Accuracy of multiple-choice (MC), True-False (TF), and Short-Answer (SA) questions evolve during iteration on MENSA with the SUMMQ_COMBO.

The cost analysis in Table 8 reveals significant differences in computational resource requirements across different approaches. While simple prompting baseline maintains the lowest cost at $0.18 per instance with minimal input and output token usage, SUMMQ_SOLO demonstrates a moderate increase in resource consumption, requiring $1.95 per instance with 0.43M input tokens and 6.97K output tokens. This reflects the additional computational overhead of our iterative refinement process compared to baseline approaches. The SUMMQ_COMBO variant shows the highest resource requirements at $14.45 per instance, consuming 3.30M input tokens and generating 24.96K output tokens, which is attributable to the collaborative multi-agent framework involving multiple summary and quiz generators, reviewers, and the iterative debate process. Despite the higher computational cost, the substantial improvements in summary quality and quiz generation accuracy demonstrated throughout our evaluation justify this investment, particularly for applications where high-quality outputs are prioritized over computational efficiency.

Table 8: Average token usage and cost (in USD) per example of different methods on MENSA with GPT-4O.

|  | Input Tokens | Output Tokens | Cost (USD) |
| --- | --- | --- | --- |
| Prompting | 0.04M | 0.23K | 0.18 |
| SUMMQ_SOLO | 0.43M | 6.97K | 1.95 |
| SUMMQ_COMBO | 3.30M | 24.96K | 14.45 |

| arXiv ID | Title |
|----------|-------|
| 2410.07095 | MLE-bench: Evaluating machine learning agents on machine learning engineering |
| 2504.13959 | AI Safety should prioritize the Future of Work |
| 2410.15522 | M-RewardBench: Evaluating Reward Models in Multilingual Settings |
| 2406.17557 | The FineWeb Datasets: Decanting the Web for the Finest Text Data at Scale |
| 2411.01493 | Sample-Efficient Alignment for LLMs |
| 2407.19056 | Benchmarking Language Agents across Multiple Applications for Office Automation |
| 2411.19943 | Critical Tokens Matter: Token-Level Contrastive Estimation Enhances LLM's Reasoning Capability |
| 2412.03679 | Evaluating Language Models as Synthetic Data Generators |
| 2412.03555 | PaliGemma 2: A Family of Versatile VLMs for Transfer |
| 2412.09871 | Byte Latent Transformer: Patches Scale Better Than Tokens |
| 2412.06559 | ProcessBench: Identifying Process Errors in Mathematical Reasoning |
| 2412.14161 | TheAgentCompany: Benchmarking LLM Agents on Consequential Real World Tasks |
| 2406.06144 | Language Models Resist Alignment: Evidence From Data Compression |
| 2410.12883 | Scaling Laws for Multilingual Language Models |
| 2410.04840 | Strong Model Collapse |
| 2406.15480 | On Giant's Shoulders: Effortless Weak to Strong by Dynamic Logits Fusion |
| 2410.08964 | Language Imbalance Driven Rewarding for Multilingual Self-improving |
| 2411.19799 | INCLUDE: Evaluating Multilingual Language Understanding with Regional Knowledge |
| 2410.07825 | Extracting and Transferring Abilities For Building Multi-lingual Ability-enhanced Large Language Models |
| 2502.17910 | Scaling LLM Pre-training with Vocabulary Curriculum |

Table 9: List of papers used for human evaluation, including arXiv ID and title.

## E    HUMAN EVALUATION

We conduct a human evaluation to assess the quality of summaries generated by different methods. It is impractical for human judges to read the entire long documents from an unfamiliar domain, so we randomly select 20 NLP papers published after June 2024, and employ five Ph.D. students who published at least one NLP paper as judges. Each judge is presented with the source document and two summaries generated by different methods, and they are asked to choose the better summary considering the following aspects, including *Informativeness*, *Coherence*, and *Factuality*. Based on the feedback from the judges that the summaries generated by SUMMQ$_{COMBO}$ (2084 words on average) are significantly longer than those from other methods (1450 words on average by O3 and 982 words on average by GPT-4O), we also include a rephrased version of SUMMQ$_{COMBO}$, denoted as SUMMQ$_{COMBOR}$, where we prompt GPT-4O to shorten the summary generated by SUMMQ$_{COMBO}$. The rephrased summaries have an average length of 1573 words, which is more comparable to the baselines. The selected papers are presented in Table 9.

## F    CASE STUDY

We present a case study in Table 10 comparing the summaries generated by GPT-4O, O3, and SUMMQ$_{COMBO}$ for the movie "Good Will Hunting". The text highlighted in red indicates details that are covered by SUMMQ$_{COMBO}$ but missed by either GPT-4O or O3. Conversely, the text highlighted in blue represents details that are included in either GPT-4O or O3 but omitted by SUMMQ$_{COMBO}$. This comparison illustrates how SUMMQ$_{COMBO}$ captures a broader range of important plot points and character developments, while also highlighting some specific elements that were overlooked. Overall, the case study demonstrates the strengths of our approach in generating comprehensive and detailed summaries.

| Model | Summary |
|---|---|
| GPT-4O | The text describes various scenes and interactions from a screenplay set in South Boston and at M.I.T. It opens with a St. Patrick's Day parade and moves into a bar where four young men-Chuckie, Will, Billy, and Morgan-share a crude story. Will Hunting, a 20-year-old with remarkable intelligence, is revealed to be working as a janitor at M.I.T. and anonymously solving complex math problems on chalkboards. Professor Lambeau discovers Will's talents and arranges for him to avoid jail by meeting with therapists, including Sean Maguire, a tough psychologist from South Boston. Will's relationship with Skylar, a Harvard student, is explored, revealing his fear of intimacy and abandonment. Sean helps Will confront his past and emotional barriers. Will ultimately decides to pursue a job at Tri-tech, but leaves a note for Sean saying he had to go ˘see about a girl,¨indicating he chose to follow Skylar to California. The story ends with Will driving away, symbolizing his decision to embrace change and new opportunities. |
| O3 | Will Hunting, a 20-year-old orphan from South Boston, spends his days drinking with blue-collar friends and nights working as an MIT janitor. When he anonymously solves an impossible math proof, Professor Gerald Lambeau tracks him down, then saves him from jail (after an assault arrest) in exchange for weekly math work and therapy. Will sabotages several therapists until he meets Lambeau's former roommate, Sean Maguire, a community-college psychologist who grew up in the same neighborhood and is still mourning his late wife. Their confrontations force Will to face the childhood abuse, class anger and fear of abandonment that make him hide behind brilliance and bravado. While Lambeau lines up lucrative jobs, Will half-heartedly dates Harvard pre-med Skylar but pushes her away when she asks him to move to California. Chuckie, his best friend, challenges Will not to waste his "lottery ticket." At last, a breakthrough with Sean convinces Will he deserves love and a future of his own choosing. Rejecting the safe corporate offer, Will leaves Boston-after gifting his therapist a note that reads, "Sorry, I had to go see about a girl"-and drives west to reunite with Skylar, finally ready to risk both his talent and his heart. |
| SUMMQCOMBO | The screenplay "Good Will Hunting" follows Will Hunting, a young man from South Boston with a troubled past and extraordinary mathematical abilities. Will spends his time with his close-knit group of friends, including Chuckie, Morgan, and Billy, in their working-class Irish neighborhood. Despite his genius, Will works as a janitor at MIT, where he anonymously solves complex mathematical problems posted by Professor Lambeau, a Field's Medal winner. The story begins with scenes from the South Boston St. Patrick's Day Parade and a humorous yet dark tale from Chuckie about his cousin Mikey Sullivan's misadventure with an injured cat. Will's life takes a significant turn after he gets into a fight with Bobby Champa, who hurts a woman with a bottle, and faces jail time. Professor Lambeau intervenes, offering Will a deal to avoid prison if he agrees to study mathematics and attend therapy sessions. Will reluctantly starts therapy with several psychologists but only begins to open up with Sean Maguire, a therapist with a similar background who has faced personal hardships, including the loss of his wife to cancer and his military service. Through their sessions, Sean helps Will confront his past traumas and fears of abandonment. Will's relationship with Skylar, a Harvard student, becomes strained as he struggles with his emotions and the idea of leaving his familiar life. Despite being offered a job at the National Security Agency, Will refuses, citing ethical concerns. He initially considers a position at Tri-Tech Laboratories but ultimately decides to follow his heart and go after Skylar, who has moved to California. The story concludes with Will leaving his old life behind, symbolized by his friends discovering his departure and Sean finding a note from Will explaining his decision to seek a future with Skylar. Chuckie gifts Will a car for his 21st birthday, which Will uses to drive away, leaving his friends and old life behind. |

Table 10: Case study on the summaries given by GPT-4O, O3, and SUMMQCOMBO about the movie "Good Will Hunting". The text in red highlights the details covered by SUMMQCOMBO but missed by either GPT-4O or O3. The text in blue highlights the details covered by either GPT-4O and O3 but missed by SUMMQCOMBO.

## G  PROMPTS

### G.1  PROMPTS OF SUMMARY GENERATORS

In this section, we present the detailed prompts used for summary generation in SUMMQ$_{\text{COMBO}}$, covering draft summary generation, refinement, aggregation, best summary selection, and voting.

---

**Summary Generators 1: Draft Summary Generation Prompt**

$\mathcal{SYS}\ \mathcal{PROMPT}$:

You are a helpful assistant tasked with summarizing long text. Summarize the following text concisely and accurately, ensuring that all key points are covered. The summary should be clear and coherent, avoiding unnecessary details or repetition. Use precise language and maintain the original meaning of the text.

---

$\mathcal{USER}\ \mathcal{PROMPT}$:

Original Text: "{Document}"

Summary:

---

**Summary Generators 2: Refine Summary Generation Prompt**

$\mathcal{SYS}\ \mathcal{PROMPT}$:

You are a helpful assistant tasked with refining summaries. Given the original text, the initial summary, feedback from the evaluator, and feedback from quiz testing, refine the summary to better capture the key points in the original text and address any shortcomings.

---

$\mathcal{USER}\ \mathcal{PROMPT}$:

Original Text: "{Document}"
Previous Summary: "{Summary}"
Reviewers Feedback: "{Summary reviewers feedback}"
Quiz Testing Feedback:
The summary could not answer the following questions correctly: "{Examinee feedback}"

Refined Summary:

---

**Summary Generators 3: Summary Aggregation Prompt**

$\mathcal{SYS}\ \mathcal{PROMPT}$:

You are an expert synthesiser. You will be given several candidate summaries of the same original text. Merge them into ONE superior summary that retains every important detail but avoids redundancy.

---

$\mathcal{USER}\ \mathcal{PROMPT}$:

Original Text: "{Document}"
Candidate Summaries: "{Candidates}"

Merged Summary:

---

**Summary Generators 4: Best Summary Selection Prompt**

$\mathcal{SYS\ PROMPT}$:

You are an expert summarisation judge. Rank the candidate summaries from best to worst according to coverage, factual accuracy and conciseness. Return the best summary.

---

$\mathcal{USER\ PROMPT}$:

Original Text: "{Document}"
Candidate Summaries: "{Candidates}"

Best Summary:

---

**Summary Generators 5: Voting Prompt**

$\mathcal{SYS\ PROMPT}$:

You are an expert and strict summarization judge. Given two summaries, determine which one is better according to coverage, factual accuracy and conciseness. ONLY Return 1 or 2, where 1 means the first summary is better and 2 means the second summary is better. If both are equally good, return 1 or 2. Reply with nothing else.

---

$\mathcal{USER\ PROMPT}$:

Original Text: "{Document}"
Candidate Summaries: "{Candidates}"

Best One (1 or 2):

---

### G.2 PROMPTS OF QUIZ GENERATORS

In this section, we present the detailed prompts for quiz generation in SUMMQ$_{\text{COMBO}}$, covering draft quiz generation, refinement, aggregation, best quiz selection, and voting.

---

**Quiz Generators 1: Draft Quiz Generation Prompt**

$\mathcal{SYS}\ \mathcal{PROMPT}$:

Multiple-choice questions:
Format:
1. Question?
A) Option 1
B) Option 2
C) Option 3
D) Option 4

True/False questions:
Format:
1. Statement. (True/False)
Short-answer question:

Format:
1. Question?

You are a helpful assistant tasked with generating questions from long text. Generate quiz questions clearly covering key points. Include: "{num of mc}" Multiple-choice questions, "{num of tf}" True/False questions, and "{num of sa}" Short-answer question.. The Question Format is as above.

---

$\mathcal{USER}\ \mathcal{PROMPT}$:

Original Text: "{Document}"

Quiz:

---

**Quiz Generators 2: Refine Quiz Generation Prompt**

$\mathcal{SYS}\ \mathcal{PROMPT}$:

You are a helpful assistant tasked with refining generated questions. Given the text, the initial generated questions, feedback from the evaluator, and feedback from quiz testing, refine the questions to ensure they cover important information clearly and avoid trivial or overly detailed content. Return "{num of mc}" Multiple-choice questions, "{num of tf}" True/False questions, and "{num of sa}" Short-answer question.

---

$\mathcal{USER}\ \mathcal{PROMPT}$:

Original Text: "{Document}"
Previous Quiz: "{Quiz}"
Reviewers Feedback: "{Quiz reviewers feedback}"
Quiz Testing Feedback:
The following questions could not be answered correctly based on the key information: "{Examinee feedback}"

Refined Quiz:

---

**Quiz Generators 3: Quiz Aggregation Prompt**

*SYS PROMPT*:

You are an expert synthesiser. You will be given several candidate generated questions of the same text. Merge them into superior questions that retains every important detail but avoids redundancy with "{num of mc}" Multiple-choice questions, "{num of tf}" True/False questions, and "{num of sa}" Short-answer question.

---

*USER PROMPT*:

Original Text: "{Document}"
Candidate Quiz Sets: "{Candidates}"

Merged Quiz:

---

**Quiz Generators 4: Best Quiz Selection Prompt**

*SYS PROMPT*:

You are an expert question generation judge. Rank the candidate questions sets from best to worst according to coverage, difficulty and clarity. Return the best question set.

---

*USER PROMPT*:

Original Text: "{Document}"
Candidate Quiz Sets: "{Candidates}"

Best Quiz:

---

**Quiz Generators 5: Voting Prompt**

*SYS PROMPT*:

You are an expert and strict question generation judge. Given two question sets, determine which one is better according to coverage, difficulty and clarity. ONLY Return 1 or 2, where 1 means the first question set is better and 2 means the second question set is better. If both are equally good, return 1 or 2. Reply with nothing else.

---

*USER PROMPT*:

Original Text: "{Document}"
Candidate Quiz Sets: "{Candidates}"

Best One (1 or 2):

---

### G.3 PROMPTS OF SUMMARY REVIEWERS

In this section, we provide the detailed prompts for summary review in SUMMQ$_{\text{COMBO}}$, including summary review annotation, merging agreed issues, and debating contested issues.

---

**Summary Reviewers 1: Annotate Summary Prompt**

$\mathcal{SYS\ PROMPT}$:

You are a strict generated summary reviewer.

1. Coverage - at least 90% of key facts needed to answer every quiz question appear.
2. Faithful - no hallucinations / contradictions.
3. Brevity - $\leq 25$ % tokens of source OR $\leq 500$ words.
4. Clarity - precise, coherent language.

If **all four** points are satisfied output exactly 'PASS' and reply with nothing else.
Otherwise list concrete problems.
For every problem output ONE line in the form:
- CATEGORY: short description
where CATEGORY is in {COVERAGE, FAITHFUL, BREVITY, CLARITY}.
If there is no problem with this category, do not output this category.

---

$\mathcal{USER\ PROMPT}$:

Original Text: "{Document}"
Quiz Questions: "{questions}"
Summary to Review: "{summary}"

Feedback:

---

**Summary Reviewers 2: Agreed Issues Merged Prompt**

$\mathcal{SYS\ PROMPT}$:

You are an expert synthesiser. You will be given several feedback for the generated summary. Merge them into ONE superior feedback that retains every important detail but avoids redundancy.

---

$\mathcal{USER\ PROMPT}$:

Original Text: "{Document}"
Summary: "{Summary}"
Candidate Feedback: "{Candidates}"

Merged Feedback:

---

**Summary Reviewers 3: Contested Issues Debate Prompt**

$\mathcal{SYS\ PROMPT}$:

You are participating in a one-turn debate about the following alleged issue in a generated summary. Reply with ONE line starting with either KEEP (keep the issue) or DISCARD (discard the issue) followed by a brief justification.

---

$\mathcal{USER\ PROMPT}$:

Original Text: "{Document}"
Quiz Questions: "{questions}"
Summary: "{Summary}"
Issues to Debate: "{Issues}"

Feedback:

### G.4   PROMPTS OF QUIZ REVIEWERS

In this section, we provide the detailed prompts for quiz review in SUMMQ$_{COMBO}$, including quiz review annotation, merging agreed issues, and debating contested issues.

---

**Quiz Reviewers 1: Annotate Quiz Prompt**

*SYS PROMPT*:

You are a strict question reviewer.
QUESTION-review rubric:

A. Coverage Distribution
1. Every *major* section / scene / argument of the chapter is addressed.
2. No cluster: questions are spread across the beginning, middle, end.

B. Cognitive Depth
• $\geq 40$ % Remember / Understand
• $\leq 20$ % Evaluate / Create

C. Format Balance
- Required counts of MC, True/False, Short-answer are respected.
- Short-answer asks for 1-2 sentences, names, dates, or concepts.
- MC: exactly 4 options, one correct; distractors plausible and non-overlapping.
- True/False: clear factual statements, no double-negatives.

D. Difficulty Gradient
• 30 % easy, 50 % medium, 20 % hard.
- Easy : answer is stated explicitly.
- Medium: answer needs light inference / synthesis.
- Hard : answer needs multi-sentence reasoning.

E. Clarity & Quality
1. Questions are grammatically correct, unambiguous, no trivia.
2. Each question targets *one* idea only.
3. No repeated facts across different questions.

If **all** points are satisfied output exactly 'PASS' and reply with nothing else.
Otherwise list concrete problems.
For every problem output ONE line in the form:
- CATEGORY: short description
where CATEGORY is in {Coverage Distribution, Cognitive Depth, Format Balance, Difficulty Gradient, Clarity & Quality }.
If there is no problem with this category, do not output it.

---

*USER PROMPT*:

Original Text: "{Document}"
Key Information: "{summary}"
Quiz to Review: "{Quiz}"

Feedback:

---

**Quiz Reviewers 2: Agreed Issues Merged Prompt**

$\mathcal{SYS\ PROMPT}$:

You are an expert synthesiser. You will be given several feedback for the generated questions. Merge them into ONE superior feedback that retains every important detail but avoids redundancy.

---

$\mathcal{USER\ PROMPT}$:

Original Text: "{Document}"
Quiz: "{Quiz}"
Candidate Feedback: "{Candidates}"

Merged Feedback:

---

**Quiz Reviewers 3: Contested Issues Debate Prompt**

$\mathcal{SYS\ PROMPT}$:

You are participating in a one-turn debate about the following alleged issue in the generated questions. Reply with ONE line starting with either KEEP (keep the issue) or DISCARD (discard the issue) followed by a brief justification.

---

$\mathcal{USER\ PROMPT}$:

Original Text: "{Document}"
Key Information: "{Summary}"
Quiz: "{Quiz}"
Issues to Debate: "{Issues}"

Feedback:

---

G.5   PROMPTS OF EXAMINEE

In this section, we present the detailed prompt for the examinee module in SUMMQ$_{\text{COMBO}}$, which is responsible for answering the generated quizzes based on the provided summaries.

---

**Examinee 1: Take Quiz Prompt**

$\mathcal{SYS\ PROMPT}$:

For every question below select the answer **in the required format**:
– Multiple-choice → return only the correct letter (A/B/C/D).
– True/False → return only the word True or False.
– Short-answer → return a short phrase or sentence taken verbatim from the text (no extra commentary).

---

$\mathcal{USER\ PROMPT}$:

Text: "{Summary}"
Questions: "{Quiz Questions}"

Return one answer per line in the same order.

---

