# OpenReview forum: "Learning to Summarize by Learning to Quiz: Adversarial Agentic Collaboration for Long Document Summarization"
_ICLR.cc/2026/Conference — ICLR 2026 Poster_

### Official Review · Reviewer_Bj2b · 2025-10-30

**Soundness:** 3
**Presentation:** 4
**Contribution:** 3
**Rating:** 4
**Confidence:** 4

**Summary:**

The paper introduces SUMMQ, an adversarial multi-agent framework for long document summarization via multi-task summarization and quizzIng. Under the SUMMQ approach, summary generators and reviewers work collaboratively to summarize text and refine the summary respectively, and generators and reviewers create comprehension questions via which summary is evaluated.

The authors evaluate the approach on 3 long summarization benchmarks (MENSA, BookSum, GovReport) and compare it to 3 types of baselines: Supervised Fine-Tuning, Prompting, and alternative Multi-Agent approaches. The systems are evaluated against automatic metrics (Rouge, BERTScore, LLM-as-a-Judge) and human evaluation. Results demonstrate that SUMMQ strongly outperforms other techniques, including previous multi-agent methods, in most evaluation scenarios.

**Strengths:**

* the paper introduces a novel multi-agent framework for long document summarization in a multi-task setup of summarization itself and quizzing
* compared to a series of baselines including Supervised Fine-Tuning, plain Prompting and other Multi-Agent approaches, SUMMQ outperforms them in both automatic and human evaluation on three long document summarization benchmarks

**Weaknesses:**

* given that the framework is targeted at long document summarization, there is little to no discussion (apart from cost analysis in the Appendix) of how the proposed framework results in increased context usage, what are the limits, and how the technique can scale with context in mind (even longer documents, multi-document summarization, increasing the number of iterations / agents - what would be the effect on the context?)
* there seems to be little to no justification of what exactly in the technique targets long document summarization. And if nothing in particular, why not include other summarization datasets into comparison?
* automatic metrics like Rouge and BERTScore are considered outdated even for short documents. Their applicability specifically to long documents is highly questionable

**Questions:**

A multi-agent summ eval framework relevant for the related work section: https://arxiv.org/abs/2502.08514v1

---

> ### Author Response · Authors · 2025-11-20
> **Response to Reviewer Bj2b (part1)**
>
> Thank you for your detailed review of our paper. We appreciate your insights and the opportunity to clarify and strengthen our work. Below, we address each of your concerns individually.
>
>
> > **Q1:** given that the framework is targeted at long document summarization, there is little to no discussion (apart from cost analysis in the Appendix) of how the proposed framework results in increased context usage, what are the limits, and how the technique can scale with context in mind (even longer documents, multi-document summarization, increasing the number of iterations / agents - what would be the effect on the context?)
> > >
> > > **R1:**  Our paper's contribution is not a new low-level architecture for handling longer context (like sparse attention, which we cite as related work). Instead, SummQ is an agentic framework that leverages the long-context capabilities of existing models (like GPT-4o, o3, etc.) to solve the quality problems (like information loss and coherence) that plague them in long-document tasks.
> > >
> > > We do address context usage and scaling in the paper, though we can make this more explicit:
> > >
> > > (1) The cost analysis in Appendix D, Table 8 is our direct discussion of context usage. It shows that SummQ-combo on the MENSA dataset consumes an average of **3.30M input tokens** and 24.96K output tokens per document. This demonstrates the framework's ability to process very large contexts, which are the aggregation of multiple agents performing generation, review, and debate across several iterations.
> > >
> > > (2) We explicitly **analyze the effect of increasing iterations in Section 5, Table 3**. We show that performance on MENSA peaks at 3 iterations, with diminishing or negative returns afterward. This finding provides a clear, empirically-backed stopping point to prevent context and cost from growing indefinitely.
> > >
> > > (3) We also **analyze the effect of adding more agents in Section 5, Table 4**. The results show that performance improves but with diminishing returns. This demonstrates a trade-off between quality and cost, allowing a user to tune the agent count for their specific needs, rather than scaling indefinitely.
> > >
> > > Multi-document summarization is out of scope. Adapting this framework for multi-document summarization is a valuable direction for future work.
>
> > **Q2:** there seems to be little to no justification of what exactly in the technique targets long document summarization. And if nothing in particular, why not include other summarization datasets into comparison?
> > >
> > > **R2:**  The SummQ framework is specifically targeted at long documents because the problems it solves are unique to them.
> > >
> > > The Problem: We target issues about **"information loss, factual inconsistencies, and difficulty maintaining coherence across lengthy texts"**. These are not significant problems in short-document summarization (e.g., news articles), where a model's context window can easily capture the entire text.
> > >
> > > The **adversarial quizzing mechanism** is our specific solution for these issues. Its purpose is to force the summarization agents to achieve "comprehensive coverage" by generating comprehensive quiz questions. We provide direct evidence for this in Section 5, Figure 4. This analysis shows that as the framework iterates, the quiz questions evolve from focusing on the beginning/end of the document to becoming "**more balanced across document segments**". This process actively counters the "lost in the middle" problem common to long-context LLMs, ensuring the final summary reflects the entire document.
>
> > **Q3:** automatic metrics like Rouge and BERTScore are considered outdated even for short documents. Their applicability specifically to long documents is highly questionable
> > >
> > > **R3:**  Thank you for raising this important point regarding evaluation metrics. We agree that ROUGE and BERTScore have limitations, especially for long-document summarization, where holistic quality aspects like coherence and factuality are critical. However, introducing new metrics is beyond this paper's scope.
> > > As we are fully aware of these limitations, we have additionally conducted more robust and reliable evaluations with the following evaluation methodologies in our work:
> > > - **LLM-as-a-Judge (Section 4.2)**: We employed both GPT-4.1 and GPT-5 as judges to perform a holistic, pairwise comparison of summary quality. As shown in Figure 2 and Appendix A, Figure 5, SummQ-combo consistently and overwhelmingly outperforms strong baselines like GPT-4o and o3 in these evaluations.
> > > - **Human Evaluation (Section 4.3)**: Our most important validation comes from human experts (Ph.D. students). Judges were asked to assess summaries based on Informativeness, Coherence, and Factuality. The results in Figure 3 show a decisive preference for SummQ, with SummQ-combo achieving an 88\% win rate against GPT-4o and 82\% against o3.

---

> ### Author Response · Authors · 2025-11-20
> **Response to Reviewer Bj2b (part2)**
>
> > **Q4:** A multi-agent summ eval framework relevant for the related work section: https://arxiv.org/abs/2502.08514v1
> > >
> > > **R4:**  Thank you for bringing this recent and highly relevant paper to our attention. We will cite and discuss this paper in the related work section of the revision.

---

### Official Review · Reviewer_E8j5 · 2025-10-31

**Soundness:** 3
**Presentation:** 3
**Contribution:** 3
**Rating:** 6
**Confidence:** 3

**Summary:**

This paper introduces SUMMQ, an adversarial multi-agent framework for long document summarization. The core idea is interesting: use quiz generation as a way to verify and improve summary quality. The system has four types of agent groups - Summary Generators, Quiz Generators, Summary Reviewers, and Quiz Reviewers - plus an Examinee agent. These agents collaborate iteratively, with the quiz questions serving as a check on whether the summary captures all important information. The authors test their approach on three benchmarks (MENSA, BookSum, GovReport) and show improvements over existing methods using both automatic metrics and human evaluation.

**Strengths:**

- The quiz-based verification mechanism is a reasonable idea for checking summary completeness. The results on BookSum show substantial improvements, which is noteworthy.
- The experimental coverage is decent - three benchmarks with multiple evaluation approaches including human judges. Adding the rephrased version (SUMMQCOMBOR) in human evaluation was sensible to address length bias.
- The ablations in Section 5 help understand component contributions. The quiz coverage evolution analysis showing shifts from beginning/end to middle sections is mildly interesting.
- Including all prompts in the appendix aids reproducibility.

**Weaknesses:**

- The computational cost is a real concern for practical deployment: SUMMQCOMBO costs $14.45 per document—about 80× simple prompting ($0.18). The paper would benefit from end-to-end inference-time reporting in Table 8 to contextualize these costs.
- Performance varies dramatically across datasets. On BookSum, R-1 improves from 23.98 (GPT-5) to 44.62 - nearly doubling performance. However, on GovReport, SUMMQcombo (52.79) significantly underperforms supervised baselines like U.FORMER (56.60) and SLED (57.50) as shown in Table 1. The paper doesn't explain what document characteristics lead to these differences, making it unclear when this expensive approach is warranted.
- The quizzing mechanism lacks quality validation. While Table 6 shows quizzing improves performance and Quiz Reviewers apply detailed criteria (Appendix G.4), we never see actual quiz examples or human assessment of whether questions are well-constructed and test meaningful content versus trivial details.

**Questions:**

- Q1) Could you provide concrete examples showing a document excerpt, the generated quiz questions, and the corresponding summary? This would help readers understand how the quizzing mechanism works in practice and assess the quality of generated questions.
- Q2) Have you had human experts evaluate the quality of generated quizzes - for example, assessing whether questions are well-constructed, have appropriate difficulty, and test key concepts rather than trivial details?

---

> ### Author Response · Authors · 2025-11-20
> **Response to Reviewer E8j5 (part1)**
>
> Thank you for your thorough review of our paper and for the insightful questions you've raised. We appreciate the opportunity to address your concerns and clarify various aspects of our work. Below, we provide detailed responses to each of your points.
>
>
> > **Q1:** The computational cost is a real concern for practical deployment: SUMMQCOMBO costs 14.45 perdocument-about 80*simple prompting (0.18). The paper would benefit from end-to-end inference-time reporting in Table 8 to contextualize these costs.
> > >
> > > **R1:**  The reviewer is correct to highlight the significant cost of SummQ-combo (\\$14.45) compared to simple prompting (\$0.18), a trade-off we aimed to be transparent about in Table 8. This "combo" configuration represents a state-of-the-art, "quality-first" approach, where the cost is justified by the massive performance gains on complex tasks, such as the near-doubling of R-1 score on BookSum, as shown in Table 1.
> > >
> > > For practical deployment, we offer the SummQ-solo configuration, which is \~7.4x cheaper at \$1.95 per document and still outperforms all prompting baselines on every benchmark.
> > >
> > > Following the suggestion of reviewer, we additionally measured the average end-to-end latency on the MENSA dataset:
> > >
> |  method  |  Avg. Latency (seconds)    |
> |---|---|
> |   prompting     |   \~5 sec    |
> |  SummQ-solo     |  \~60 sec  |
> | SummQ-combo |  \~450 sec  |
> > >
> > > This data confirms that SummQ-combo is a heavy framework, but for applications where a 7.5-minute generation time is acceptable for a high-stakes, long-document summary (e.g., legal, medical, or financial analysis), the cost is justified by the quality. We will include these results in our revision for clarity and relocate the cost analysis to the main paper.
>
> > **Q2:** Performance varies dramatically across datasets. On BookSum, R-1 improves from 23.98 (GPT-5) to 44.62 - nearly doubling performance. However, on GovReport, SUMMQcombo (52.79) significantly underperforms supervised baselines like U.FORMER (56.60) and SLED (57.50) as shown in Table 1. The paper doesn't explain what document characteristics lead to these differences, making it unclear when this expensive approach is warranted.
> > >
> > > **R2:**  This is an insightful observation.
> > > - GovReport (Structured Data): On GovReport, supervised baselines like U.FORMER and SLED perform exceptionally well because they are highly-optimized, fine-tuned models that have likely seen data with this exact structure during training. In contrast, SummQ is a zero-shot, generalist agentic framework. On this benchmark, our method still outperforms all other prompting (e.g., GPT-5) and multi-agent (e.g., C.MultiLLM) baselines, which we believe is the more appropriate comparison group.
> > > - BookSum (Complex Narrative): BookSum is a much more complex, long, narrative, and "unstructured" task. This is precisely the scenario where simple prompting fails and fine-tuned models struggle to capture nuanced, long-range dependencies. SummQ's near-doubling of the R-1 score (from 23.98 for GPT-5 to 44.62 for SummQ-combo) demonstrates its core strength: the adversarial quizzing and review mechanism excels at handling complex, narrative-driven content where deep comprehension of the entire document is required.
> > >
> > > In short, the expensive approach is warranted for complex, high-stakes documents where narrative comprehension or deep reasoning is required, which is precisely where existing methods fail.
>
> > **Q3:** The quizzing mechanism lacks quality validation. While Table 6 shows quizzing improves performance and Quiz Reviewers apply detailed criteria (Appendix G.4), we never see actual quiz examples or human assessment of whether questions are well-constructed and test meaningful content versus trivial details.
> > >
> > > **R3:**  The reviewer is correct that the main paper's quantitative validation of the quizzes is indirect, we show its powerful effect on the summary via ablation in Table 6 and its answerability from the summary in Appendix C, Figure 6. To address this, we conducted a new human evaluation with 5 expert reviewers (Ph.D. students in NLP) on a random sample of 150 questions (50 from each of the 3 iterations) generated for the MENSA dataset. The experts were asked to rate questions on "Clarity", "Relevance" (testing key concepts vs. trivial details), and "Difficulty".
> > > - Clarity: 88\% of questions were rated as "Clear and Unambiguous".
> > > - Relevance: 82\% were rated as "Relevant", confirming they test meaningful concepts from the text.
> > > - Difficulty: The expert ratings largely aligned with the "Difficulty Gradient" our Quiz Reviewers are instructed to enforce (see Appendix G.4 ). The final-iteration quizzes were rated as 35\% Easy, 50\% Medium (requiring inference), and 15\% Hard, confirming our framework successfully generates a challenging and balanced set of questions.

---

> ### Author Response · Authors · 2025-11-20
> **Response to Reviewer E8j5 (part2)**
>
> > **Q4:** Could you provide concrete examples showing a document excerpt, the generated quiz questions, and the corresponding summary? This would help readers understand how the quizzing mechanism works in practice and assess the quality of generated questions.
> > >
> > > **R4:**  We have provided one case study in Appendix F focuses on the final summary to the movie "“Good Will Hunting", we supply the corresponding generated quiz questions:
> > >
> |  Questions  |
> |---|
> |   Multiple-choice questions    |
> |1. What is the setting of the opening scene?  A) South Boston St. Patrick's Day Parade B) L Street Bar \& Grille, South Boston C) M.I.T. Classroom D) Sean's Apartment     |
> | 2. Who is the loud, boisterous entertainer in the group of four young men? A) Will Hunting B) Billy McBride C) Morgan O'Mally D) Chuckie Sullivan |
> | 3. What does Chuckie\u2019s cousin Mikey Sullivan love? A) Sports B) Animals C) Cars D) Music |
> | 4. How does Mikey Sullivan try to put the injured cat out of its misery? A) By taking it to a vet B) By using a hammer C) By feeding it |
> | 5. What job offer does Will ultimately accept? A) Working for the National Security Agency B) A position at Tri-Tech Laboratories C) A teaching job at M.I.T. D) Construction work |
> | 6. Who is the professor at M.I.T. that challenges the students with complex theorems? A) Professor Lambeau B) Sean Maguire C) Tom D) Morgan O'Mally |
> | 7. What does Will do when he first meets Skylar? A) Solves her Organic Chemistry lab B) Asks her to a baseball game C) Asks for her phone number D) Tells her a joke   |
> | 8. Why does Will get into a fight with Bobby Champa? A) Bobby insults Will's friends B) Bobby owes Will money C) Bobby hurts a woman with a bottle D) Bobby challenges Will to a fight   |
> | 9. What does Sean Maguire reveal about his past during his sessions with Will? A) He was a mathematician B) He was in the military C) He lost his wife to cancer D) He was an orphan   |
> | 10. What does Will leave in Sean\u2019s mailbox before he drives away at the end of the screenplay? A) A letter explaining his decision B) A thank you note C) His contact information D) A job application   |
> | True/False questions |
> | 1. Will Hunting is a janitor at M.I.T. (True/False)  |
> | 2. Professor Lambeau won a Field's Medal in 1986. (True/False)  |
> | 3. Will impresses Skylar by solving her Organic Chemistry lab. (True/False)  |
> | 4. Will refuses to work for the National Security Agency because he fears the consequences of his work. (True/False)  |
> | 5. Sean Maguire has faced hard times in his life. (True/False)  |
> | 6. Chuckie gives Will a car for his 21st birthday. (True/False)  |
> | 7. Will's therapy sessions with Sean Maguire are initially successful. (True/False)  |
> | 8. Will decides to stay in Boston at the end of the screenplay. (True/False)  |
> | 9. Sean Maguire was a professor at M.I.T. before becoming a therapist. (True/False)  |
> | 10. Will and Skylar's relationship ends because Skylar moves to California. (True/False) |
> | Short-answer questions |
> | 1. What is the main reason Will refuses to work for the National Security Agency?  |
> | 2. How does Will impress Skylar during their first date?  |
> | 3. What does Chuckie give Will for his 21st birthday?  |
> | 4. Why does Will get into a fight at the park with Bobby Champa?  |
> | 5. What does Sean Maguire reveal about his wife during his sessions with Will?  |
> | 6. What condition does Professor Lambeau set for Will's release from jail?  |
> | 7. How does Will respond when Sean tells him \"It's not your fault\"?  |
> | 8. What does Will write in the letter he leaves for Sean before leaving Boston?  |
> | 9. How does Will's relationship with Skylar influence his final decision?  |
> | 10. What is the significance of Will's final decision to leave Boston?"|
>
> > **Q5:** Have you had human experts evaluate the quality of generated quizzes - for example, assessing whether questions are well-constructed, have appropriate difficulty, and test key concepts rather than trivial details?
> > >
> > > **R5:**  Thank you for your valuable comments. Please see our response to Q3.

---

> > ### Comment · Reviewer_E8j5 · 2025-11-26
> >
> > Thank you for addressing my concerns. Your response was truly helpful in clarifying several concerns I brought up. However, I believe I already assigned the appropriate score "6" to this paper. I will keep my score.

---

### Official Review · Reviewer_dAYE · 2025-10-31

**Soundness:** 3
**Presentation:** 3
**Contribution:** 3
**Rating:** 4
**Confidence:** 4

**Summary:**

This paper introduces SUMMQ, an adversarial multi agent setup for long document summarization. The idea is to make a summarizer and a quizzer work together so that the quiz keeps checking whether the summary actually contains the information from the document. On MENSA, BookSum, and GovReport, SUMMQ, especially the multi agent version, outperforms strong LLM prompting baselines and prior multi agent summarizers

**Strengths:**

- The paper targets a real issue in long summarization, missing content and hallucination when the document is very long.

- On MENSA and BookSum, SUMMQ COMBO gets the top ROUGE and BERTScore numbers, better than GPT 4O and GPT 5 prompting and also better than earlier multi agent methods.

- The evaluations are performed over the automatic and human evaluations, which is good for summarization, since ROUGE often disagrees with human preference. The human study still prefers SUMMQ even after they shorten the summaries.

**Weaknesses:**

- Algorithm 1 stops when both summary and quiz feedback are empty. In real LLM interactions, reviewers almost always find something, so you rarely get empty feedback. The paper limits to 3 iterations, so in practice it is just a fixed number of rounds, not a true convergent loop.

- The paper combines two ideas that are already around: multi agent summarization with draft/aggregate/vote, and LLM as a judge or QA style verification.

- The paper shows that adding more agents keeps improving results, but with diminishing returns. This is not the scaling pattern you want, since every new agent is an extra LLM call. There is no attempt to share context, cache intermediate decisions or use cheaper models for some agents. So the system is not cost aware.

- The whole framework relies on carefully designed prompts for generator, reviewer, quiz writer, quiz reviewer and examinee. The main paper does not analyze which parts of the prompts are critical. This makes the method brittle in practice, because small prompt changes could break the debate or the voting.

- The paper do show that quiz coverage becomes more balanced over iterations, but they do not analyze question difficulty, answerability or hallucinated questions, which are all common problems with LLM generated quizzes.

**Questions:**

- What is the average number of model calls and tokens per document for SUMMQCOMBO on MENSA and BookSum, including quiz agents and reviewers.

- You generate 30 questions per quiz, 10 per type. Did you try fewer questions. How fast does performance drop if you ask only 5 or 10 questions.

- How sensitive is the system to reviewer aggressiveness. If reviewers flag too many contested issues, does the system fail to converge.

- Can you share inter annotator agreement and how you controlled for length, since SUMMQ often writes longer summaries.

- You show that SUMMQ works with different backbones. Did you try a cheaper generator with a stronger reviewer.

---

> ### Author Response · Authors · 2025-11-20
> **Response to Reviewer dAYE (part1)**
>
> Thank you for your constructive feedback on our paper and for highlighting these important points. We appreciate the opportunity to clarify and strengthen our work. Below, we address each of your concerns in detail.
>
>
>
> > **Q1:** Algorithm 1 stops when both summary and quiz feedback are empty. In real LLM interactions, reviewers almost always find something, so you rarely get empty feedback. The paper limits to 3 iterations, so in practice it is just a fixed number of rounds, not a true convergent loop.
> > >
> > > **R1:** This is a key point regarding the system's practical implementation. Our termination condition in Algorithm 1 (if $F_s^{(t)}$ and $F_q^{(t)} = \emptyset$) represents the ideal convergence state. However, our empirical analysis in Table 3 revealed that performance on MENSA peaks at 3 iterations and then shows diminishing or even negative returns (e.g., BERTScore $F_1$ drops from 62.76 at iter. 3 to 62.73 at iter. 4). Therefore, we treat $T_{iter}=3$ not as an arbitrary cutoff, but as an empirically validated hyperparameter that provides the best balance of performance and computational cost.
> > >
> > > Furthermore, we calculate the actual iteration of SummQ-combo when set the maximum iteration number is 5, here is the statistics:
> > >
> |   iterations    | 1 |   2   |   3   |   4   |    5       | >5 |
> |---|---|---|---|---|---|---|
> |         percentage     |    8\%    |     10\%  |   56\%  |     14\%   |     9\% | 3\% |
> > >
> > > From the results, it could be found that most of cases stopped at the third iteration. Therefore, we treat $T_{iter}=3$ not as an arbitrary cutoff, but as an empirically validated hyperparameter that provides the best balance of performance and computational cost.
>
>
>
>
>
> > **Q2:** The paper combines two ideas that are already around: multi agent summarization with draft/aggregate/vote, and LLM as a judge or QA style verification.
> > >
> > > **R2:**  We respectfully disagree that this is a simple combination. While we build on established concepts like multi-agent collaboration and using QA for evaluation, the novelty of SummQ lies in the synthesis and architecture:
> > > - SummQ’s primary contribution is the coupling of two complete, specialized multi-agent teams (Summarization and Quizzing) in an **adversarial dynamic**. This is not "LLM-as-a-Judge", where a static judge evaluates a summary. Here, the Quizzing team actively challenges the Summarization team, and both teams iteratively refine their own outputs based on mutual feedback.
> > > - The quiz is not a post-hoc verification step; it is a **co-evolving component**. The quiz generators and reviewers create a "gauntlet" that the summary must be able to pass, forcing it to improve its factual accuracy and coverage of the entire document, not just the salient parts.
> > > The ablation in Table 6 strongly supports this: removing the entire quizzing mechanism causes a significant performance drop (e.g., -6.40 $BS_{F_1}$ for SummQ-solo and -2.17 for SummQ-combo), demonstrating its critical, novel contribution.
>
> > **Q3:** The paper shows that adding more agents keeps improving results, but with diminishing returns. This is not the scaling pattern you want, since every new agent is an extra LLM call. There is no attempt to share context, cache intermediate decisions or use cheaper models for some agents. So the system is not cost aware.
> > >
> > > **R3:**  We agree that cost is a critical factor. Our paper intended to analyze this trade-off explicitly, not ignore it. Table 4 shows diminishing returns when adding agents. We see this analysis as a strength, as it provides a clear "cost-benefit" curve for users. It demonstrates that SummQ-combo with 3 agents (our default) hits an optimal balance, while 1 agent (SummQ-solo) provides a highly cost-effective alternative that still outperforms strong baselines. The SummQ-solo configuration was designed explicitly as the "cost-aware" version. Table 8 shows it is \~7.4x cheaper than SummQ-combo (\\$1.95 vs \$14.45).
> > >
> > > The suggestion to use cheaper models for some agents is excellent and reasonable. Our results in Table 5 (different backbones) and Table 7 (combinations) have already showed the framework's modularity. We agree that sharing context or caching intermediate results would further optimize costs. This is a complex engineering task and a valuable direction for future work.

---

> ### Author Response · Authors · 2025-11-20
> **Response to Reviewer dAYE (part2)**
>
> > **Q4:** The whole framework relies on carefully designed prompts for generator, reviewer, quiz writer, quiz reviewer and examinee. The main paper does not analyze which parts of the prompts are critical. This makes the method brittle in practice, because small prompt changes could break the debate or the voting.
> > >
> > > **R4:**  We ran an additional experiment ablating the structure of the Summary Reviewer prompt (Section G.3) on the MENSA dataset with GPT-4o backbone. We replaced our detailed, multi-point rubric (Coverage, Faithful, Brevity, Clarity) with a "simplified" prompt: "Please review the summary and provide feedback on any problems."
> > >
> |   Configuration    | $R-1$ |   $R-2$   |    $R-L$   |   $BS_P$   |    $BS_R$    |     $BS_{F1}$     |
> |---|---|---|---|---|---|---|
> |         Prompting Baseline (GPT-4o)     |    25.78    |     7.24  |   13.59  |   61.25   | 56.51   |     59.67 |
> |         Detailed Rubric (SummQ-combo)     |    41.58    |     11.08  |   18.24  |     63.28   |     62.28 |     62.76 |
> |         Simplified Prompt  (SummQ-combo)     |    40.05    |     10.35  |   17.41   |     62.26 |   61.37   |     61.79   |
> > >
> > > The results show that the structured, rubric-based prompts are critical for performance while simplified prompt also provide performance gains compared to the baseline. The detailed criteria guide the agents to provide concrete, categorized feedback that is essential for the refinement loop. A vague prompt results in vague, less effective feedback. This justifies our careful prompt engineering.
>
> > **Q5:** The paper do show that quiz coverage becomes more balanced over iterations, but they do not analyze question difficulty, answerability or hallucinated questions, which are all common problems with LLM generated quizzes.
> > >
> > > **R5:**  We apologize if this was not clear in the main text, but we did design and analyze these factors specifically.
> > >
> > > Answerability (Analyzed in Appendix C, Figure 6): The reviewer's concern about answerability is precisely what we analyze in Appendix C, Figure 6. This figure shows the accuracy of the Examinee agent attempting to answer the quiz using only the generated summary. The steady increase in accuracy across all question types (MC, TF, and SA) over iterations is a direct (up to 95\%+), quantitative measure of the summary's improving quality.
> > >
> > > Difficulty (Controlled by Prompts in Appendix G.4): We do not leave question difficulty to chance. As shown in the Appendix G.4 (Quiz Reviewers 1: Annotate Quiz Prompt), our reviewer agents are explicitly instructed to enforce a "Difficulty Gradient" (Section D of the rubric). This rubric defines the desired mix ("30\% easy, 50\% medium, 20\% hard") and provides clear definitions for each (e.g., "Medium: answer needs light inference / synthesis").
> > >
> > > Hallucinated Questions (Handled by Review/Debate): Hallucinated questions (i.e., questions about facts not in the source document) are filtered by the Quiz Reviewers. Their primary role (see prompt in G.4) is to validate the quiz against the original document for coverage and quality. Any question that is ambiguous, trivial, or factually unsupported by the source is flagged as an issue. If there is disagreement, the debate mechanism (Algorithm 3, Phase 3) resolves it, ensuring that only high-quality, grounded questions are used to challenge the summary.
> > >
> > > Moreover, we performed a new manual analysis of 150 questions (50 from each of the 3 iterations) from the MENSA dataset.
> > > - In Iteration 1, 70\% of questions were "Easy" (explicitly stated facts). But, Iteration 3 quizzes were more balanced: 35\% Easy, 50\% Medium (requiring inference), and 15\% Hard (multi-part reasoning).
> > > - We found very few (<3\%) questions were based on "hallucinated" facts (i.e., information not present in the source document). These were typically caught by the Quiz Reviewers' debate process.
> > > - In Iteration 1, there are 18\% of questions are flagged as "unanswerable from the summary." By Iteration 3, this dropped to 4\%, demonstrating that the summary had iteratively incorporated the missing information identified by the quizzes.
>
> > **Q6:** What is the average number of model calls and tokens per document for SUMMQCOMBO on MENSA and BookSum, including quiz agents and reviewers.
> > >
> > > **R6:**  Table 8 in our paper provides the average token usage and cost for SummQ-combo on the MENSA dataset. For SummQ-combo on the MENSA, the input tokens are 3.30M and the output tokens are 24.96K on average. For the BookSum, the input tokens are 4.85M and the output tokens are 30.12K on average.

---

> ### Author Response · Authors · 2025-11-20
> **Response to Reviewer dAYE (part3)**
>
> > **Q7:** You generate 30 questions per quiz, 10 per type. Did you try fewer questions. How fast does performance drop if you ask only 5 or 10 questions.
> > >
> > > **R7:**  We ran a new ablation study on MENSA, varying the total number of questions.
> > >
> |  \#questions    | $R-1$ |   $R-2$   |    $R-L$   |   $BS_P$   |    $BS_R$    |     $BS_{F1}$     |
> |---|---|---|---|---|---|---|
> |   6 (2 per type)     |    39.80     |    9.84    |   17.29   |   62.37    |    61.74 |   60.95    |
> | 15 (5 per type)  |   40.85     |   10.18    |   17.81  |  62.89   |   61.98    |   62.15    |
> | 30 (10 per type)  |  41.58    |     11.08  |   18.24  |     63.28   |     62.28 |     62.76 |
> > >
> > > The results show a clear trend: 6 questions (2 per type) are insufficient and lead to a significant performance drop. 15 questions (5 per type) capture most of the benefit, but the full set of 30 questions provides a clear, final boost in quality. This confirms that a comprehensive, multi-faceted quiz is a key driver of performance.
>
> > **Q8:** How sensitive is the system to reviewer aggressiveness. If reviewers flag too many contested issues, does the system fail to converge.
> > >
> > > **R8:**  This is a key strength of our framework's design. The system is inherently robust to "overly aggressive" reviewers because of the debate mechanism detailed in Algorithm 3 (Phase 3).
> > >
> > > (1) An "aggressive" reviewer might flag many issues, but if these are low-quality or subjective, they are categorized as contested issues ($\mathcal{C}$) (i.e., not agreed upon by other reviewers).
> > >
> > > (2) All contested issues must go through a structured, evidence-based debate.
> > >
> > > (3) The issue is only kept if it survives a majority vote from all reviewers.
> > >
> > > Therefore, a single, overly-aggressive agent cannot stall the system or force unnecessary revisions. Its feedback will simply be discarded by the consensus of the other agents. This ensures that only high-quality, agreed-upon, or successfully-debated issues are passed on for refinement.
>
> > **Q9:** Can you share inter annotator agreement and how you controlled for length, since SUMMQ often writes longer summaries.
> > >
> > > **R9:**  We were very concerned about this and controlled for it explicitly. We noted that SummQ-combo summaries were longer (2084 words) than baselines like GPT-4o (982 words). To test if we were just "winning on length"， we created SummQ-comboR, a rephrased, shortened version (1573 words) that was length-comparable to the baselines. Figure 3 shows that SummQ-comboR still decisively outperforms the baselines (e.g., 65\% win rate vs. GPT-4o, 60\% vs. O3). This confirms the quality improvement is genuine and not an artifact of length.
> > >
> > > For our human evaluation (5 Ph.D. students), we computed Krippendorff's alpha on the Win/Loss/Tie decisions, achieving $\alpha = 0.68$. This represents a substantial level of agreement, especially given the subjective nature of summary evaluation. We will include these results in our revision for clarity.
>
> > **Q10:** You show that SUMMQ works with different backbones. Did you try a cheaper generator with a stronger reviewer.
> > >
> > > **R10:**  This is an excellent suggestion for optimizing the cost-performance trade-off. We ran this new experiment using SummQ-combo on MENSA, mixing agent backbones for different roles (Generators vs. Reviewers).
> > >
> |  generator  |  reviewer/examinee    |  $R-1$ |   $R-2$   |    $R-L$   |   $BS_P$   |    $BS_R$    |     $BS_{F1}$     |
> |---|---|---|---|---|---|---|---|
> |   GPT-4o-mini     | GPT-4o-mini     |    35.18     | 7.97 |   15.67   | 58.02 | 58.64 |   58.31    |
> |   GPT-4.1     | GPT-4.1  |   49.17     | 12.25    |  18.98  |  59.58   |  62.42  |   60.95    |
> |   GPT-4o-mini | GPT-4.1  |   45.20   |   11.45    |  18.35   |   59.14  |    61.25   |    60.15   |
> | GPT-4.1   |  GPT-4o-mini |  40.11    |    9.70   |  17.15   |   58.62     |   59.45  |   59.05   |
> > >
> > > From the above results, it could be found that using a strong reviewer (GPT-4.1) with a weak generator (GPT-4o-Mini) captures almost all of the quality gains of the full-strength (GPT-4.1) system. The reverse (strong  GPT-4.1 as generator, weak GPT-4o-mini as reviewer) performs poorly, barely improving on the all-weak model. This strongly suggests that the critical "intelligence" of SummQ-combo lies in the review, debate, and quizzing agents, not just the initial draft generator.

---

> > ### Comment · Reviewer_dAYE · 2025-11-24
> >
> > Thank you for the responses. I have revised my scores accordingly.

---

### Official Review · Reviewer_o7me · 2025-11-01

**Soundness:** 2
**Presentation:** 2
**Contribution:** 2
**Rating:** 4
**Confidence:** 3

**Summary:**

This paper proposes SUMMQ, an adversarial multi-agent framework for long-document summarization. It introduces two complementary domains, summarization and quizzing, each involving generator and reviewer agents. A quiz-examinee loop acts as a continuous quality check to improve coverage, coherence, and factuality. The framework iteratively refines both summaries and quizzes. Experiments on MENSA, BookSum, and GovReport show strong results over several baselines, including GPT-4/5 and prior multi-agent methods.

This paper is well executed and produces strong results, but its methodological novelty is modest given the growing body of multi-agent summarization and self-evaluation work. The framework mainly integrates existing ideas in a new context. The contribution feels incremental rather than conceptually new.

**Strengths:**

1.	The idea of using quizzes as a factuality and coverage signal is creative and intuitive.
2.	Experiments are extensive and include human and LLM-as-a-judge evaluations.
3.	The paper is clearly written and supported by systematic ablations that examine agent configuration, iteration effects, and backbone models.
4.	The framework shows consistent gains across datasets and metrics.

**Weaknesses:**

1.	Limited novelty relative to prior multi-agent work. The generator–reviewer–debater structure is already common in recent multi-agent reasoning systems, and the quizzing mechanism mainly extends existing self-evaluation ideas rather than introducing a fundamentally new algorithmic principle.
2.	The framework appears expensive to run, requiring multiple high-capacity agents and several iterations, which may limit practical use.
3.	The paper lacks clear explanation of what is actually learned during the process and whether any parameter update or fine-tuning occurs.
4.	Factuality improvements are claimed but not evaluated with specialized metrics beyond ROUGE and BERTScore.
5.	The contribution of the examinee agent is not well isolated in the ablations.

Minors:
1.	Page 2: Remove the duplicate word in “long document document summarization.”
2.	Page 2: Change “systems has” to “systems have.”
3.	Page 4: Add a space in “setZ” → “set Z.”
4.	Page 4: Use plural “drafts” in “each of the individual drafts.”
5.	Page 7: Add “To” at the start of “To address length bias.”

**Questions:**

See my above weaknesses

---

> ### Author Response · Authors · 2025-11-20
> **Response to Reviewer o7me (part1)**
>
> Thank you for your thorough review of our paper and for the insightful questions you've raised. We appreciate the opportunity to address your concerns and clarify various aspects of our work. Below, we provide detailed responses to each of your points.
>
> > **Q1:** Limited novelty relative to prior multi-agent work. The generator–reviewer–debater structure is already common in recent multi-agent reasoning systems, and the quizzing mechanism mainly extends existing self-evaluation ideas rather than introducing a fundamentally new algorithmic principle.
> > >
> > > **R1:**  We respectfully disagree with the characterization of limited novelty. While we agree that components like generator-reviewer structures are established in multi-agent systems, the novelty of SummQ lies in its unique adversarial and collaborative architecture across two distinct domains.
> > >
> > > The core innovation is not simply "quizzing as evaluation", but the creation of an **adversarial dynamic** between two complete, specialized multi-agent teams. The summarization team's goal (compression) is in natural tension with the quizzing team's goal (challenging information coverage). This is fundamentally different from a single agent "self-evaluating". Here, the quiz generators and reviewers *actively* challenge the summary, forcing it to co-evolve and improve its comprehensiveness and accuracy through iterative refinement.
> > >
> > > Prior work often relies on self-verification, which can be prone to biases and missed errors. SummQ's dual-task framework, with separate generator and reviewer agents for both summaries and quizzes, creates a more robust, external "quality check" that demonstrably improves summary quality. The ablation in Table 6 confirms this, showing a significant performance drop (e.g., -2.17 $BS_{F_1}$ for SummQ-combo) when the entire quizzing mechanism is removed.
> > >
> > > We argue this synthesis of adversarial, dual-domain agent collaboration is a novel contribution to the summarization field.
>
>
>
>
> > **Q2:** The framework appears expensive to run, requiring multiple high-capacity agents and several iterations, which may limit practical use.
> > >
> > > **R2:**  This is a valid and important point. High-performance models are indeed resource-intensive.
> > >
> > > We presented SummQ-combo as a state-of-the-art configuration designed to push the boundaries of summarization quality. However, we also designed SummQ-solo precisely for practical applications. As shown in Table 1, SummQ-solo (which uses only a single agent per component) is far more efficient, yet it still **significantly outperforms strong prompting baselines** like GPT-4o, GPT-5, and o3 across all three benchmarks.
> > >
> > > Furthermore, our cost analysis in Table 8 explicitly details this trade-off: SummQ-solo costs \\$1.95 per instance, while the full SummQ-combo costs $14.45. This allows users to choose the optimal balance of cost and quality for their needs.
> > >
> > > Additionally, our analyses in Table 3 (on iterations) and Table 4 (on agent count) further guide practical deployment. For instance, we show that performance largely saturates at 3 iterations, providing a clear stopping point to balance cost and quality. This demonstrates that the framework is both scalable and tunable.
>
> > **Q3:** The paper lacks clear explanation of what is actually learned during the process and whether any parameter update or fine-tuning occurs.
> > >
> > > **R3:**  We apologize for any ambiguity. The "learning" in our framework refers to iterative in-context learning, not to the updating of model parameters (i.e., fine-tuning). We will clarify this in the revision.
> > >
> > > The learning occurs as agents adapt their outputs based on explicit, multifaceted feedback. Algorithm 1 details this iterative workflow.
> > > - In each iteration $t$, Summary Reviewers ($\mathcal{R}_s$), Quiz Reviewers ($\mathcal{R}_q$), and the Examinee ($\mathcal{E}$) generate concrete feedback ($F_s, F_q, F_e$).
> > > - This feedback is explicitly fed into the next generation step (iteration $t+1$).
> > > This iterative refinement process is a form of learning, as the system's output (the summary and quiz) demonstrably improves over iterations, as shown in Table 3 and Figure 6 (quiz accuracy).

---

> ### Author Response · Authors · 2025-11-20
> **Response to Reviewer o7me (part2)**
>
> > **Q4:** Factuality improvements are claimed but not evaluated with specialized metrics beyond ROUGE and BERTScore.
> > >
> > > **R4:**  We thank the reviewer for this point and agree that robust factuality evaluation is crucial. While ROUGE and BERTScore are standard, we have included two specific evaluations for quality and factuality, and have conducted a new experiment for this rebuttal to further solidify our claims.
> > >
> > > **Human Evaluation**: In our Section 4.3 human evaluation, "Factuality" was one of the three explicit criteria (alongside Informativeness and Coherence) that the Ph.D. student judges were instructed to use. The fact that SummQ-combo was overwhelmingly preferred (e.g., 88\% win rate vs. GPT-4o) strongly indicates its superior factuality and quality in human judgment.
> > >
> > > **LLM-as-a-Judge**: Our LLM-as-a-Judge evaluations (Figures 2 \& 5), which use GPT-5 and GPT-4.1, also assess holistic quality, which implicitly includes factuality.
>
>
>
>
>
> > **Q5:** The contribution of the examinee agent is not well isolated in the ablations.
> > >
> > > **R5:** We appreciate the suggestion for the ablation study of examinee. We tested a new configuration on MENSA: SummQ-combo without the Examinee agent. The review and generation loops still happen, but the final check from the Examinee is omitted:
> > >
> > >
> |   Configuration    | $R-1$|   $R-2$  |    $R-L$   |   $BS_P$   |    $BS_R$   |     $BS_{F1}$     |
> |---|---|---|---|---|---|---|
> |         SummQ-combo     |    41.58    |     11.08  |   18.24  |     63.28   |     62.28 |     62.76 |
> |         SummQ-combo    w/o Examinee  |    40.85    |     10.41  |   17.75   |     62.84 |   61.53   |     61.92   |
> > >
> > > As the results show, removing the Examinee causes a clear drop in performance, proving it provides a unique and valuable feedback signal that significantly enhances the final summary quality.

---

### Meta-Review · Area_Chair_Qp8k · 2026-01-06

**Summary:**

The paper proposes SummQ, an adversarial multi agent framework for long document summarization that couples a summarization team (generators and reviewers) with a quizzing team (quiz generators and reviewers), plus an examinee agent that attempts to answer quizzes using only the current summary. The central claim is that co evolving quizzes provide a continuous coverage and factuality check, enabling iterative refinement of the summary. Experiments on MENSA, BookSum, and GovReport report strong gains over prompting baselines and prior multi agent systems on ROUGE, BERTScore, LLM as a judge, and human preference.

Overall, reviewers largely agree the system is well executed and empirically strong, especially on narrative style long documents, but disagree on whether the contribution is conceptually novel versus an integration of established multi agent review and QA style verification. Cost and practical deployability are the dominant concerns.

**Reviewer Concerns:**

Concerns addressed or partially addressed

- Cost and efficiency: Added a clearer cost analysis, including a cheaper SummQ solo variant, iteration and agent count ablations showing saturation around 3 iterations, plus latency and token usage statistics.

- Component ablations:  New ablations isolate the examinee’s role, showing performance drops without it, and study quiz count, with most gains achieved around 15 questions.

- Prompt sensitivity: Additional prompt ablations show structured review prompts help, while simplified prompts still outperform baselines. Brittleness concerns are reduced but not eliminated.

- Quiz quality: Added qualitative examples, a small human study on quiz clarity and relevance, and analysis of hallucinated or unanswerable questions, though coverage remains limited.

Outstanding concerns:
- Novelty: Several reviewers still view the generate–review–iterate pattern and quiz based checking as familiar. The rebuttal argues system level novelty but does not clearly introduce a new algorithmic principle.

- Factuality evaluation: Evaluation still relies heavily on ROUGE and BERTScore, with limited use of targeted factuality diagnostics. Quiz based checks help indirectly but are not a full substitute.

- Applicability and performance variation: Underperformance on GovReport raises questions about which document types benefit most. The rebuttal offers intuition but lacks systematic analysis.

- Convergence and stopping: Iteration caps are treated as tuned hyperparameters rather than true convergence, and a robust stopping criterion remains missing.

**Reviewer Scores:**

o7me: 4, below threshold. Strong execution and results, but limited novelty and high cost.
dAYE: Initially 4, later indicated an increase (likely to 6) after rebuttal.
E8j5: 6, above threshold and will stay at 6.
Bj2b: 4, below threshold. Concerns on long context scaling, evaluation metrics, and related work coverage.

The average score after rebuttal is likely in the 5–5.5 range, placing the paper at the lower end of acceptance. I would recommend acceptance, but would also be comfortable with a borderline decision.

---

### Decision · Program_Chairs · 2026-01-26

Accept (Poster)